# Understanding Spatiotemporal Development of Human Settlement in Hurricane-prone Areas on U.S. Atlantic and Gulf Coasts using Nighttime Remote Sensing

Xiao Huang[1], Cuizhen Wang[1], Junyu Lu[2]

[1]Department of Geography, University of South Carolina, Columbia, 29208, U.S.A
[2]School of Community Resources and Development, Arizona State University, Phoenix, 85004, U.S.A

**Correspondence to:** Xiao Huang (xh1@email.sc.edu)

**Abstract.** Hurricanes, as one of the most devastating natural hazards, have posed great threats to people in coastal areas. A better understanding of the spatiotemporal dynamics of human settlement in hurricane-prone areas largely benefits sustainable development. This study uses the DMSP/OLS nighttime light (NTL) data to examine the human settlement development in areas with different levels of hurricane proneness from 1992 to 2013. The DMSP/OLS NTL data from six satellites were intercalibrated and desaturated with AVHRR and MODIS optical imagery to derive the vegetation-adjusted NTL urban index (VANUI), a popular index that quantifies human settlement intensity. The derived VANUI time series was examined with the Mann-Kendall test and Theil-Sen test to identify significant spatiotemporal trends. To link the VANUI product to hurricane impacts, four hurricane-prone zones were extracted to represent different levels of hurricane proneness. Aside from geographic division, a wind-speed weighted track density function was developed and applied to historical North Atlantic Basin originated storm tracks to better categorize the four levels of hurricane proneness. Spatiotemporal patterns of human settlement in the four zones were finally analyzed. The results clearly exhibit a north-south and inland-coastal discrepancy of human settlement dynamics. This study also reveals that both the zonal extent and zonal increase rate of human settlement positively correlate with hurricane proneness levels. The intensified human settlement in high hurricane-exposure zones deserves further attention for coastal resilience.

**1 Introduction**

Hurricane, a specific type of tropical cyclone with wind speed of 74 miles per hour (mph) or higher, is one of the most devastating natural hazards in the world and is recurring more frequently than ever in coastal areas (Vecchi and Knutson, 2018). Based on Saffir-Simpson Hurricane Scale, a hurricane is categorized in five levels by its wind speed: 74-95 mph as

Category 1; 96-110 mph as Category 2; 111-129 mph as Category 3; 130-156 mph as Category 4; above 157 mph as Category 5. Hurricanes threating the conterminous United States are mostly originated from the North Atlantic Basin that includes the North Atlantic Ocean, the Caribbean Sea and the Gulf of Mexico, and Eastern Pacific Basin that covers Northeastern Pacific (east of 140ºW and north of the equator) (Goldenberg et al., 2001). Historically, more hurricanes from the North Atlantic Basin have landed on the U.S territories, dramatically affecting people living in Gulf coasts and Atlantic coasts. While the Eastern

Pacific Basin originated storms occasionally visited the southwestern conterminous U.S, by the time they landed they usually degraded to tropical cyclones due to the long travel distance and cold water in coastal California (Chenoweth and Landsea, 2004).

Atlantic hurricane season usually runs from June $1^{st}$ to November $30^{th}$, during which the North Atlantic Basin exhibits significantly intensified tropical cyclone activity and gives rise to many devastating hurricanes landing the coasts. In 2016,

Hurricane Mathew, a Category 5 (the highest category) hurricane, claimed a total of 34 direct deaths in the U.S. In 2017, Hurricane Harvey in the Gulf coast caused a total of 125 billion dollars of damage, ranking the second costliest hurricanes in the U.S. In the same year, Hurricane Irma in the Atlantic coast caused a total of 50 billion dollars of damage, ranking the fifth costliest hurricanes in the U.S ("Costliest U.S. tropical cyclones tables updated", 2018). In 2018, the third year in a consecutive series (2016-2018) of above-average damaging Atlantic hurricanes, there were 15 named tropical storms, eight of which

became hurricanes, including two major hurricanes. Hurricane Florence for example, as a major hurricane in 2018, has caused severe economic damage to North Carolina ($22 billion), South Carolina ($5.5 billion) and Virginia ($1 billion) (Krupa, 2018). The widespread storm surge and extensive floods from extreme rainfall largely crippled public infrastructures and impacted all segments of society. A noticeable increase in the number of hurricanes from the North Atlantic Basin since the late 1980s has been observed (Vecchi and Knutson, 2018). Even though it is partly due to improved monitoring (Villarini et al., 2011),

the increased intensity and duration of these hazards have posed great threats to people residing in the U.S. Atlantic and Gulf Coasts (Landsea et al., 2010).

Despite these threats, the U.S. southeastern region has experienced significant population growth in recent decades. The population in Florida, North Carolina, and South Carolina, for instance, has increased by 61.2%, 43.6%, and 54.3% respectively since 1990 (U.S Census Bureau, 2018). The densely populated coastal areas are receiving higher threats than ever

(Crosset, 2005). In these hurricane-prone areas, a better understanding of the temporal and spatial dynamics of human settlement is needed for better damage assessment and sustainable urban planning.

Satellite-based observations have been widely applied in investigating urban dynamics as remote sensing provides spatially explicit information of the urbanization process. Extensive application has been made utilizing multispectral sensors that record

the reflectance of ground features to categorize different land covers, thus allowing the delineation of urban extent (Xu, 2008; Zha, 2003). This type of remotely sensed imagery, however, relies on the reflective characteristics of all land objects on the ground, thus lacking the perspective on human activities. In comparison, satellite-derived nighttime light (NTL) data provides a unique and direct observation of human settlement via night lights (Ceola et al., 2014; Ceola et al., 2015). Natural land covers are distinctively dark in NTL imagery. Nighttime remote sensing has been increasingly used for analyzing socioeconomic dynamics and urbanization process at national and regional levels (Elvidge et al., 1997; Ghosh et al., 2010), thanks to their light-only sensitivity, large spatial coverage (Imhoff et al., 1997), easiness to acquire (Lu et al., 2008) and consistency over a long term (Elvidge et al., 1999).

Among all the satellite-derived NTL products, the NTL data obtained by Operational Linescan System (OLS) via the U.S. Air Force Defense Meteorological Satellite Program (DMSP), hereafter referred to as DMSP/OLS NTL, is the most commonly used due to its long-time span (more details in next section). Extensive attempts have been made to harvest the NTL observations from DMSP/OLS in applications including urban expansion and decay (Lu et al., 2018), settlement dynamics (Elvidge et al., 1999; Yu et al., 2014), socioeconomic development (Doll et al., 2000) and energy consumption (Chand et al., 2009). Recent studies enhanced the NTL products by fusing DMSP/OLS NTL data with natural land cover characteristics such as the Normalized Difference Vegetation Index (NDVI) to reduce the light saturation problem. This fusion greatly increased the potential of DMSP/OLS in discriminating against the human settlement structures (Lin et al., 2014; Liu et al., 2015). The improved DMSP/OLS NTL product serves as a valuable resource for monitoring large-coverage and long-term urbanization dynamics.

The goal of this paper is to illustrate the usage of DMSP/OLS NTL data to monitor the urbanization process and hurricane impacts on the U.S. Atlantic and Gulf coasts using nighttime artificial lights as a proxy. Hurricane-prone areas were first derived by calculating the track density from historical storm tracks in the North Atlantic Basin. An intercalibrated DMSP/OLS NTL time series was built in a yearly interval. Assisted with the NDVI data, the Vegetation Adjusted NTL Urban Index (VANUI) was used to characterize human settlement intensity in the study area. After that, a trend analysis was conducted to identify areas with a significant increase in human settlement intensity in different zones, in which the potential hurricane impacts were statistically evaluated. The spatiotemporal changes of human settlement revealed from nighttime remote sensing in hurricane-prone zones provide valuable information to evaluate the damage and to support the decision making of urban development.

## 2 Intercalibration and desaturation of DMSP/OLS NTL series

Due to the absence of on-board calibration and intercalibration, the annual DMSP/OLS NTL composites derived from multiple satellites in a span of 22 years were not comparable directly (Li and Zhou, 2017; Liu et al., 2012). This lack of continuity and comparability has posed great challenges in DMSP/OLS NTL based trend analysis (Tan, 2016). Elvidge et al. (2009) designed a three-step framework to intercalibrate the DMSP/OLS NTL composites. Those three steps are: 1) selecting a reference region;

2) selecting a reference satellite year; 3) performing a $2^{nd}$-order polynomial regression against the NTL reference data. This simple framework has been proven efficient in reducing discrepancies in digital number (DN) values of the DMSP/OLS NTL time series (Pandey et al., 2013) and has been adopted in many studies (Liu and Leung, 2015; Huang et al., 2016).

Another notable limitation of DMSP/OLS NTL is the saturation of luminosity in the 6-bit (DN in a range of 0-63) imagery (Letu et al., 2010). To retrieve the heterogeneity in areas with high intensity of human settlement, numerous attempts have been made to mitigate the saturation effect. A commonly used vegetation index, NDVI, is a useful indicator to reduce the saturation effect in DMSP/OLS data. Its practicality has been confirmed by many studies (Zhou et al., 2014; Liu et al., 2015). Lu et al. (2008) proposed a human settlement index (HSI) by merging normalized DMSP/OLS NTL data with the maximum NDVI in growing season derived from Moderate Resolution Imaging Spectroradiometer (MODIS). HSI has been proved rather efficient for settlement mapping in several testing sites in southeastern China. Zhang et al. (2013) develop a vegetation-adjusted NTL urban index (VANUI), which captures the inverse correlation between vegetation and luminosity. This simple index efficiently reveals the heterogeneity in regions with saturated DN values, which has been recognized by Shao and Liu (2014). Following the original design of NDVI that characterizes the inverse relationship between the near-infrared band and red band in vegetation, Zhang et al. (2015) designed a normalized difference urban index (NDUI) that characterizes the inverse relationship between vegetation and luminosity in a similar way. NDUI was evaluated in five testing sites in the U.S and proved to be effective in desaturating DN values in DMSP/OLS.

In this study, the intercalibration of DMSP/OLS data follows the method proposed by Elvidge et al. (2009) and the desaturation of DMSP/OLS data is achieved by using VANUI (Zhang et al., 2013).

## 3 Datasets

### 3.1 Historical storm tracks

The historical storm tracks were retrieved from International Best Track Archive for Climate Stewardship (IBTrACS), hosted by NOAA (https://www.ncdc.noaa.gov/ibtracs/). The IBTrACS provides a globally best track dataset by merging storm information from multiple centers into one product. As the majority of the storms on the conterminous U.S are formed in the North Atlantic Basin (Fig. 1), we only examined the storms from the North Atlantic Basin along the U.S. Atlantic and Gulf Coasts. A total of 655 storm tracks containing 18,929 line segments (with an attribute of wind speed) were used in this study.

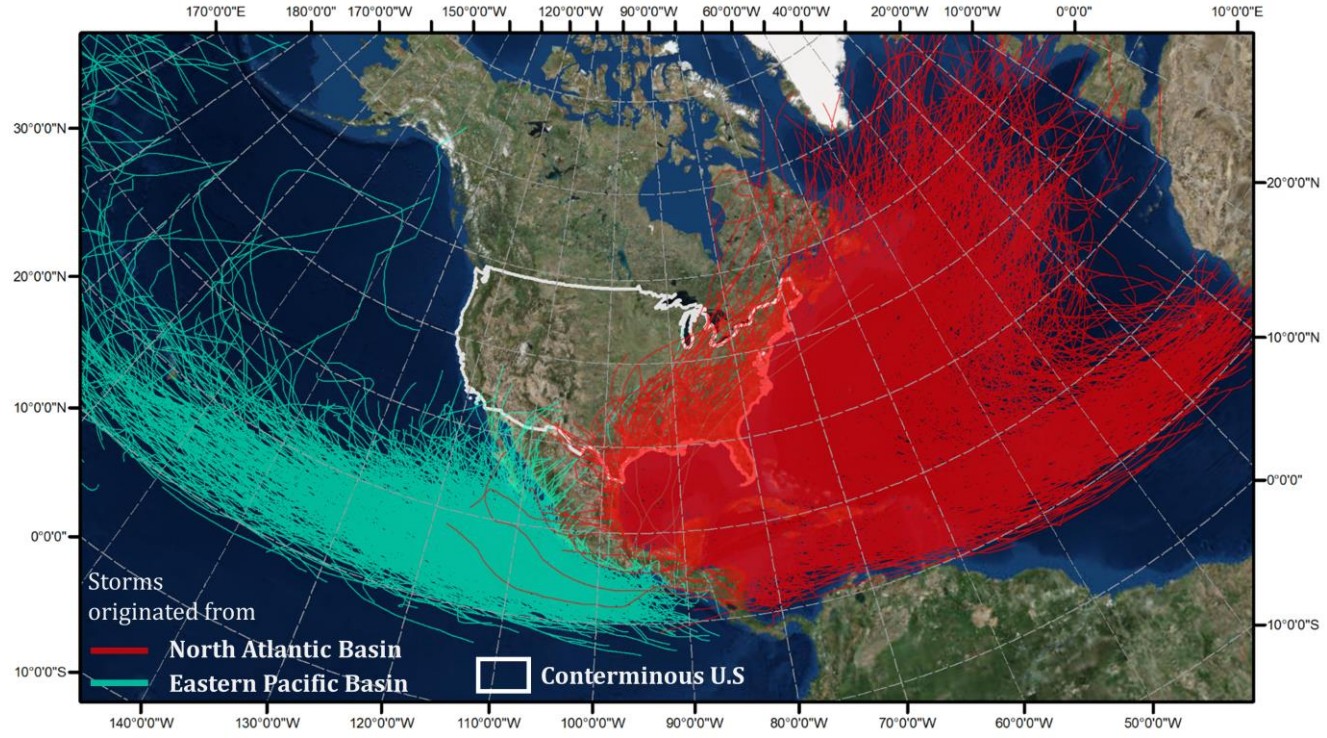

**Figure 1: Historical storm tracks from the North Atlantic Basin (in red) and from the Eastern Pacific Basin (in green).**

### 3.2 DMSP/OLS NTL series and NDVI series

The DMSP/OLS satellites are operated by U.S Air Force (USAF) and are composed of six satellites (F10, F12, F14, F15, F16, and F18) in the period of 1992-2013. With a 3,000 km orbit swath, they acquired the OLS imagery from $-65°$ to $65°$ in latitude at a nominal resolution of 30 arc-second (around 1 km at the Equator) (NOAA Earth Observation Group, 2018). The temporal coverages of the six satellites are summarized in Table 1.

**Table 1.**
DMSP/OLS Satellites and overlays in corresponding years.

| | Satellites | | | | | |
|---|---|---|---|---|---|---|
| Year | F10 | F12 | F14 | F15 | F16 | F18 |
| 1992 | F101992 | | | | | |
| 1993 | F101993 | | | | | |
| 1994 | **F101994** | **F121994** | | | | |
| 1995 | | F121995 | | | | |
| 1996 | | F121996 | | | | |
| 1997 | | **F121997** | **F141997** | | | |
| 1998 | | **F121998** | **F141998** | | | |
| 1999 | | **F121999** | **F141999** | | | |
| 2000 | | | **F142000** | **F152000** | | |
| 2001 | | | **F142001** | **F152001** | | |
| 2002 | | | **F142002** | **F152002** | | |

| Year | | | | |
|------|---------|---------|---------|---------|
| 2003 | **F142003** | **F152003** | | |
| 2004 | | **F152004** | **F162004** | |
| 2005 | | **F152005** | **F162005** | |
| 2006 | | **F152006** | **F162006** | |
| 2007 | | **F152007** | **F162007** | |
| 2008 | | | F162008 | |
| 2009 | | | F162009 | |
| 2010 | | | | F182010 |
| 2011 | | | | F182011 |
| 2012 | | | | F182012 |
| 2013 | | | | F182013 |

*Note.* Bold terms indicate the years with two satellites available in a given year.

The DMSP/OLS NTL products used in this study are the version 4 Stable Lights series in a 22-year span (1992-2013). The DMSP/OLS NTL data were obtained from the National Centers for Environmental Information website (https://ngdc.noaa.gov/eog/dmsp/downloadV4composites.html). The version 4 DMSP/OLS Stable Lights product has already

excluded sunlit, glare, moonlit, cloud coverage, and lighting. Ephemeral events such as wildfires also have been discarded. In this study, one composite each year in the conterminous U.S was produced from each satellite. When two satellites were available in certain years, a combined composite in this year was derived using the method described in Section 4.2. All DMSP/OLS NTL images were resampled to the 1 km pixel size.

In the same period of 1992-2013, the NDVI products in the conterminous U.S from two satellite sensors were used in this

study: Advanced Very High Resolution Radiometer (AVHRR) and Moderate Resolution Imaging Spectroradiometer (MODIS). NDVI series from AVHRR and MODIS span from 1992-2005 and 2003-2013, respectively. These two products were further calibrated in three overlaying years: 2003, 2004, and 2005 (described in Section 5.1) to increase data comparability. AVHRR NDVI series is the annual maximum value composite (MVC) with 1 km pixel size, provided by United States Geological Survey Earth Resources Observation and Science (USGS/EROS)

(https://phenology.cr.usgs.gov/get_data_1km.php). A number of preprocessing steps have been performed in this product to remove noises, which includes removal of spurious spikes, temporal smoothing, and interpolation. MODIS NDVI series was derived from Oak Ridge National Laboratory Distributed Active Archive Center (ORNL DAAC) (https://daac.ornl.gov/). The data were generated from Terra MOD13Q1 and Aqua MYD13Q1 products and have been smoothed and gap-filled with a spatial resolution of 250 m (Spruce et al., 2016). To be comparable with AVHRR NDVI, the annual MVC product was derived

from the MODIS NDVI series by selecting the maximum NDVI value in each year. It was also resampled to 1 km pixel size. Water bodies contained in both datasets were masked out using MODIS MOD44W product.

## 4 Methods

### 4.1 Delineation of hurricane-prone zones

The delineation of hurricane-prone zones is based on the retrieved 655 storms from the North Atlantic Basin landed on the conterminous U.S. An area with higher hits of historical storms is expected to be more hurricane-prone. We assume a generally positive relationship between the wind intensity of a storm and its impact. At a given location $(i,j)$, a circular neighborhood (R) centered at this location was assigned. For all line segments of storm tracks falling in this neighborhood, the storm track density was calculated as the line density of all segments weighted by their wind speeds:

$$\rho_{i,j} = \sum_{r \in R} L_{i,j}^r \times W_{i,j}^r, \tag{1}$$

where $\rho_{i,j}$ denotes the weighted line density at the location $(i,j)$. $L_{i,j}^r$ and $W_{i,j}^r$ denote the length of a line segment $r$ and corresponding wind speed, respectively. The radius of R is set as 100 km in this study.

The storm track density was then normalized to a range of [0,1], with a higher value indicating higher hurricane proneness. To simplify the process for zonal analysis, we categorized the normalized storm track density into four zones from low to high hurricane proneness: Zone 4 (0-0.2), Zone 3 (0.2-0.5), Zone 2 (0.5-0.7) and Zone 1 (0.7-1.0).

### 4.2 Intercalibration (DMSP/OLS NTL series; NDVI series) and VANUI calculation

We adopted the Elvidge et al. (2009) procedure to intercalibrate the DMSP/OLS NTL time series. Serving as the reference site (Fig. 3a), the geographic area of metropolitan Los Angeles and City of San Diego, CA maintains high conformity of NTL values throughout the 22-year period (Kyba et al., 2017; Hsu et al., 2015), which satisfies the "pseudo invariant" rule for calibration site selection (Elvidge et al., 2009). The year 2007 (satellite F16) has been commonly selected as the reference year in many studies (Yi et al., 2014; Ma et al., 2014). Therefore, we extracted the DMSP/OLS NTL data in this year at the same site as our reference. With all lit pixels (DN >0) in the reference site, a second-order regression model was performed to calibrate the NTL data in each year:

$$DN_{n,cal} = c + b \times DN_n + a \times DN_n^2, \tag{2}$$

where $DN_{n,cal}$ is the calibrated DN value in year $n$, $DN_n$ is the original DN value in year $n$ and $a, b$ and $c$ are the coefficients. The non-lit pixels (DN=0) are not calibrated.

As shown in Table 1, two DMSP/OLS NTL data layers are available in overlapping years. For lit pixels (DN>0 in both years), the calibrated DN values in this year are calculated as the average of two calibrated data sets. The value of a pixel remains 0 if its original DN value in any year is 0. Finally, the calibrated DMSP/OLS NTL images were normalized ($DN_{nor}$) to [0,1].

Similarly, the annual maximal NDVI ($NDVI^{MVC}$) products from AVHRR ($NDVI_{AVHRR}^{MVC}$ from 1992 to 2005) and MODIS ($NDVI_{MODIS}^{MVC}$ from 2003 to 2013) were intercalibrated to maintain the continuity and comparability in $NDVI^{MVC}$ annual series. Stratified sampling was applied to pixels with NDVI value above 0.1 to ensure that land covers in different NDVI ranges were

equally sampled. Thirty thousand samples were collected within four hurricane-prone zones in years 2003, 2004 and 2005, respectively. It has been reported that MODIS maintains higher spectral sensitivity than AVHRR (Tucker et al., 2005). Here, a linear regression was applied to correct AVHRR $NDVI^{MVC}$ to MODIS $NDVI^{MVC}$:

$$NDVI^{MVC}_{MODIS} = \alpha \times NDVI^{MVC}_{AVHRR} + \beta, \tag{3}$$

where $\alpha$ and $\beta$ are regression coefficients.

The calibrated $NDVI^{MVC}_{AVHRR}$ series from 1992-2002 was merged with $NDVI^{MVC}_{MODIS}$ from 2003-2013 to form a 22-year NDVI MVC series ($NDVI^{MVC}_{cal}$). Negative NDVI values are usually associated with non-living environments such as water bodies and NDVI values above 1 are not meaningful. Therefore, we limited all NDVI values in the $NDVI^{MVC}_{cal}$ series to a range of 0 to 1.

Finally, with the normalized DMSP/OLS NTL and the calibrated NDVI data series, the VANUI series was extracted (Zhang

et al. 2013):

$$VANUI = (1 - NDVI^{MVC}_{cal}) \times DN_{nor}, \tag{4}$$

where $DN_{nor}$ denotes the normalized DMSP/OLS NTL value and $NDVI^{MVC}_{cal}$ denotes the calibrated $NDVI^{MVC}$ value. The VANUI has a range of [0,1]. In general, a higher proportion of human settlements in a pixel leads to higher NTL and lower NDVI, both contributing to a higher value of VANUI. Therefore, the VANUI serves as a proxy of the intensity of human

settlement.

**4.3 Trend analysis of human settlement**

The VANUI series in a 22-year span shed light on the spatiotemporal development of the human settlement. We performed the trend analysis by applying the Mann-Kendall test (Mann, 1945) coupled with the Theil-Sen slope estimator (Sen, 1968). The Mann-Kendall test statistically assesses if there is a significant monotonic upward or downward trend in a time series.

Given the 22-year VANUI series, the Mann-Kendall test first computes $S$ statistics (Mann, 1945):

$$S = \sum_{k=1}^{n-1} \sum_{j=k+1}^{n} sgn(x_j - x_k), \tag{5}$$

where $n$ denotes the total number of observations in a series (22 in this study), $x_j$ and $x_k$ are the data values at different points, i.e., the VANUI in different years in this study. $sgn(x_j - x_k)$ denotes an indicator that takes on the values 1, 0, or -1 respectively according to the signs of $(x_j - x_k)$. The variance of $S$ ($Var_S$) is further computed as:

$$Var_S = \frac{1}{18}\left[n(n-1)(2n+5) - \sum_{p=1}^{g} t_p\,(t_p-1)(2t_p+5)\right], \tag{6}$$

where $g$ denotes the number of tied groups and $t_p$ denotes the number of observations in the $p$th group. Finally, a $Z$ value is calculated as:

$$Z = \begin{cases} \dfrac{S-1}{\sqrt{Var_S}}, & S > 0 \\ 0, & S = 0 \\ \dfrac{S+1}{\sqrt{Var_S}}, & S < 0 \end{cases} \tag{7}$$

The $Z$ value in Eq.7 represents the monotonic tendency of a time series. A positive $Z$ indicates an increasing trend, while a negative $Z$ indicates a decreasing one. A stable trend exists when the value of Z equals 0. The absolute value of $Z$ indicates the intensity of the trend. The significance of $Z$ was examined through a two-tail test with a significance level of $\alpha = 0.05$. If a significant trend exists, the Theil-Sen slope estimator was further applied to estimate its slope. As a non-parametric indicator, it has low sensitiveness to outliers and high robustness in short-term series and has been widely applied in remote sensing fields (de Jong et al., 2011; Fernandes and Leblanc, 2005). Given a VANUI time series, the slope at any point $i$ ($Q_i$) can be calculated as:

$$Q_i = \frac{x_j - x_k}{j - k}, i = 1,2,3, \dots N, j > k \tag{8}$$

The Theil-Sen slope ($Q_{med}$) is the median of all $Q_i$ values in the time series. It indicates the steepness (change rate) of a certain trend. Therefore, pixels with high $Q_{med}$ values represent a rapid increase in human settlement intensity during the investigated time period.

With the 22-year VANUI image series, clusters of geographic areas in the study region with a significant increase of human settlement were extracted. The summed slope per unit in a cluster represented the rapidness of human settlement growth in the 22 years. The spatiotemporal patterns of this growth in different hurricane-prone zones were further analyzed.

## 5 Results and discussion

### 5.1 Hurricane-prone zones

The 655 storms from the North Atlantic Basin landed on the conterminous U.S (mostly along Atlantic and Gulf coasts) are presented in Fig. 2a. The derived wind speed-weighted track density in the study area is presented in Fig. 2b. Based on the density levels, we divided the track density map into four hurricane-prone zones that represent different levels of hurricane impacts: the highest impacts in Zone 1 and lowest in Zone4. The study area contains all U.S. states covered in the hurricane-prone zones (Fig. 2c): Maine, Massachusetts, New Jersey, New York, North Carolina, New Hampshire, Pennsylvania, Rhode Island, Tennessee, Texas, Maryland, Alabama, Arkansas, Connecticut, Delaware, DC, Florida, Georgia, Kentucky, Louisiana, Mississippi, South Carolina, Vermont, Virginia and West Virginia. Some of these states such as Florida, Texas and North Carolina are well recognized as fast-growing in both population and economy in recent years (Milesi et al., 2003; Klotzbach et al., 2018), leading to higher threats and recovery costs from hurricanes.

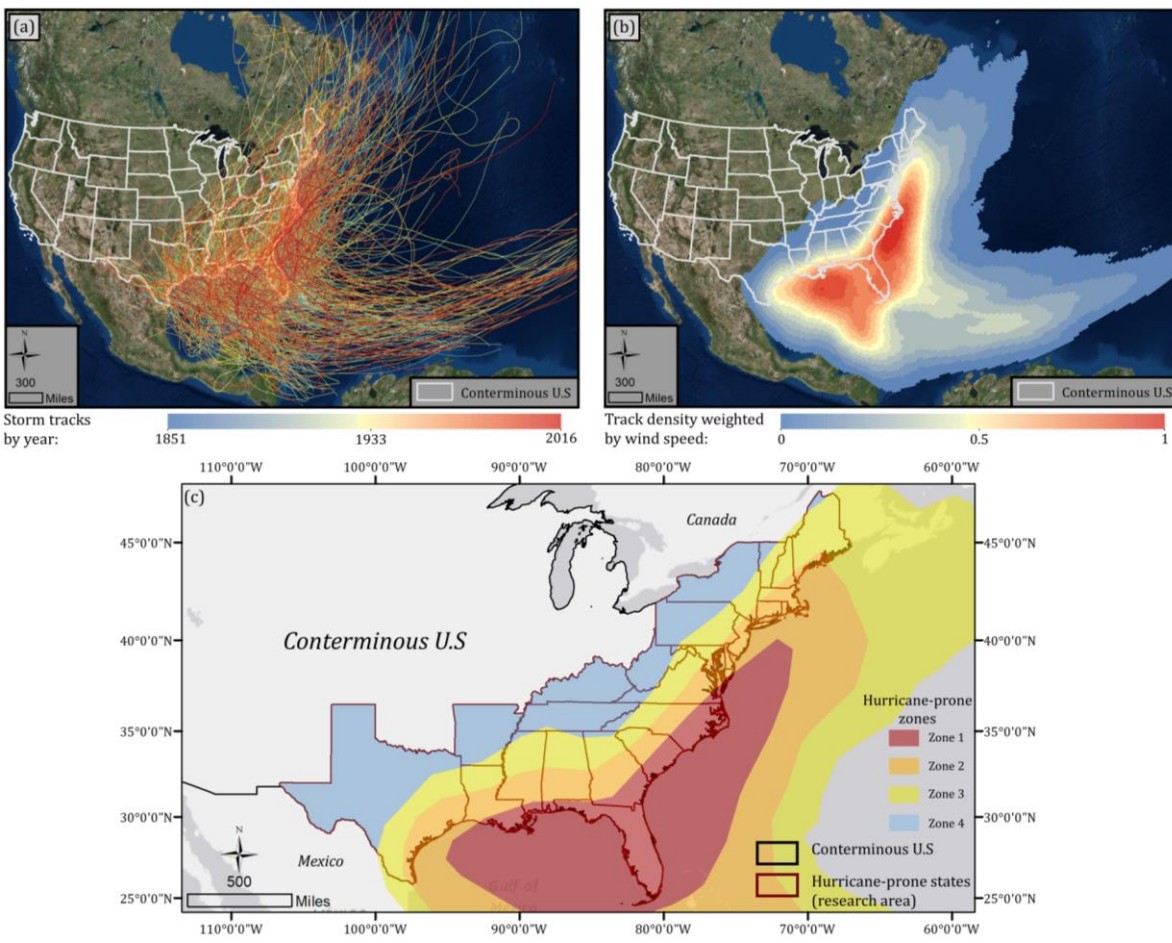

**Figure 2: (a) Historical storm tracks from the North Atlantic Basin; (b) Normalized storm track density-weighted by wind speed; (c) Hurricane-prone zones: Zone 4 (with track density 0 – 0.2), Zone 3 (0.2 – 0.5), Zone 2 (0.5 - 0.7) and Zone 1 (0.7 – 1.0).**

### 5.2 Intercalibration results of DMSP/OLS NTL series and NDVI series

5    The reference site for intercalibration is composed of an urban stripe from Los Angeles to San Diego, CA in the southwest end of the United States (Fig. 3a). Agreeing with Elvidge et al. (2009), the histograms of all NTL images in this area exhibit a sharp, bimodal distribution (urban vs. non-urban) with limited temporal variation. This confirms that it is a valid reference site for intercalibration of NTL images. Among the three example scatterplots between the NTL data in three years and the F162007 reference, the F162006 data show the highest agreement with the reference as they were acquired by the same satellite (Fig.

10  3b1). The F101992 data (Fig. 3b2) exhibit less agreement due to its different satellite origin and a long time interval from 2007. However, an $R^2$ of 0.949 still warrants a decent agreement for calibration. Fig. 3b3 demonstrates the necessity of a second-order regression instead of a linear one. The regression equations and intercalibration coefficients for all years are listed in Table 2.

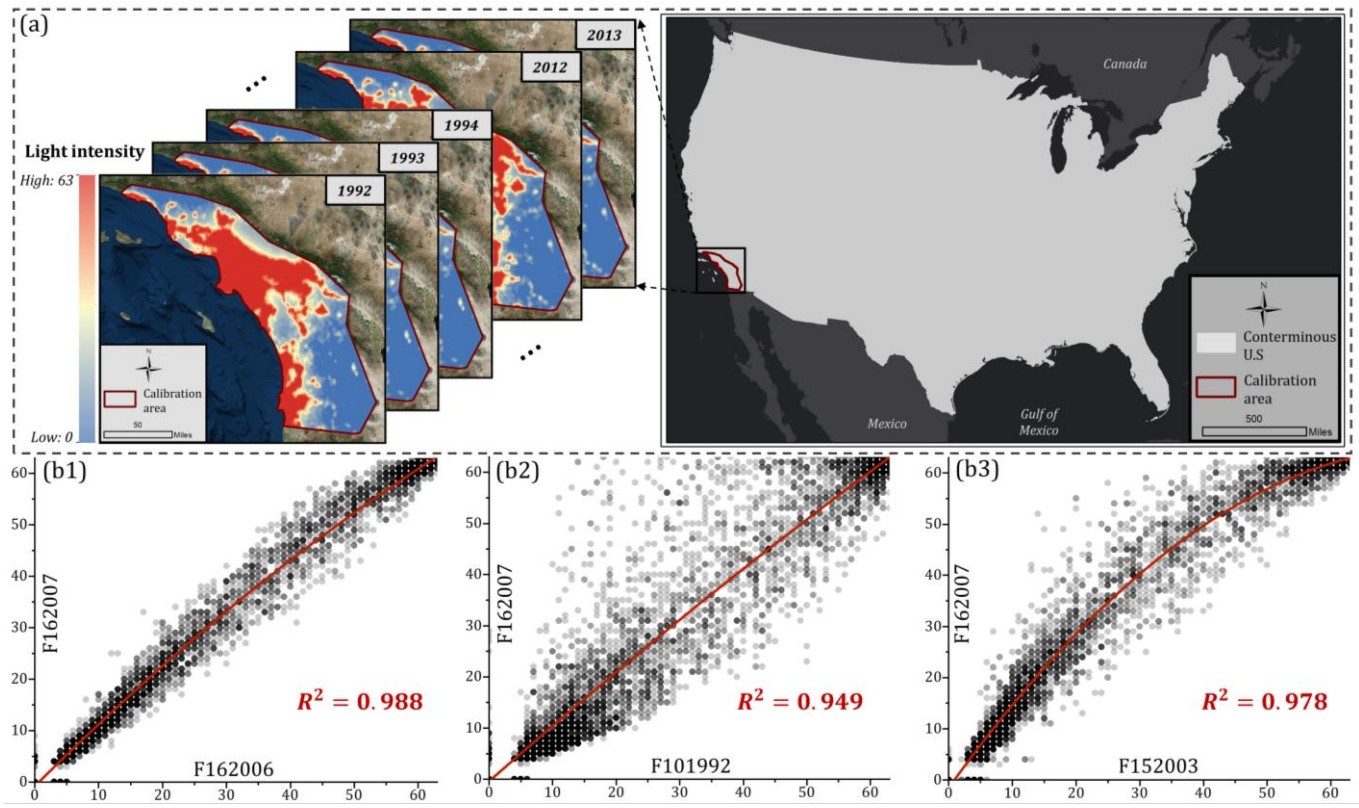

**Figure 3: (a) DMSP/OLS NTL intercalibration in L.A metropolitan and City of San Diego; (b1) Correlation between F162006 and reference year F162007; (b2) Correlation between F101992 and reference year F162007; (b3) Correlation between F152003 and reference year F162007.**

**Table 2.**

DMSP/OLS NTL intercalibration coefficients

| Satellite | Year | $c$ | $b$ | $a$ | $R^2$ |
|-----------|------|---------|--------|---------|-------|
| F10 | 1992 | -0.3712 | 1.0953 | -0.0015 | 0.949 |
| F10 | 1993 | -1.4938 | 1.4753 | -0.0072 | 0.955 |
| F10 | 1994 | -0.9394 | 1.4923 | -0.0077 | 0.951 |
| F12 | 1994 | -0.0430 | 1.2057 | -0.0033 | 0.954 |
| F12 | 1995 | -0.6145 | 1.2354 | -0.0037 | 0.955 |
| F12 | 1996 | -0.3298 | 1.2840 | -0.0045 | 0.945 |
| F12 | 1997 | 0.0253 | 1.1669 | -0.0029 | 0.934 |
| F12 | 1998 | 0.2550 | 1.0688 | -0.0013 | 0.949 |
| F12 | 1999 | -0.3859 | 0.9984 | -0.0001 | 0.967 |
| F14 | 1997 | 0.1852 | 1.5516 | -0.0090 | 0.936 |
| F14 | 1998 | -0.1074 | 1.4379 | -0.0071 | 0.959 |
| F14 | 1999 | -0.5429 | 1.4508 | -0.0070 | 0.967 |
| F14 | 2000 | -0.4461 | 1.3396 | -0.0053 | 0.969 |
| F14 | 2001 | -0.2633 | 1.4454 | -0.0071 | 0.974 |
| F14 | 2002 | 0.3598 | 1.3926 | -0.0065 | 0.961 |
| F14 | 2003 | -0.0390 | 1.3677 | -0.0059 | 0.979 |
| F15 | 2000 | -1.0303 | 1.1837 | -0.0027 | 0.967 |

| | | | | | |
|---|---|---|---|---|---|
| F15 | 2001 | -0.8264 | 1.1821 | -0.0027 | 0.977 |
| F15 | 2002 | -0.6087 | 1.1485 | -0.0022 | 0.981 |
| F15 | 2003 | -1.2553 | 1.6417 | -0.0099 | 0.978 |
| F15 | 2004 | -0.6269 | 1.6067 | -0.0095 | 0.981 |
| F15 | 2005 | -0.8131 | 1.5621 | -0.0086 | 0.980 |
| F15 | 2006 | -0.4824 | 1.3515 | -0.0054 | 0.989 |
| F15 | 2007 | -0.4583 | 1.4299 | -0.0066 | 0.983 |
| F16 | 2004 | -0.0440 | 1.3285 | -0.0053 | 0.968 |
| F16 | 2005 | -1.0392 | 1.5749 | -0.0088 | 0.986 |
| F16 | 2006 | -0.6923 | 1.2201 | -0.0033 | 0.988 |
| **F16** | **2007** | **0.0000** | **1.0000** | **0.0000** | **1.000** |
| F16 | 2008 | -0.0982 | 0.9931 | 0.0002 | 0.989 |
| F16 | 2009 | -0.1023 | 1.1478 | -0.0024 | 0.979 |
| F18 | 2010 | 0.1369 | 0.7924 | 0.0030 | 0.972 |
| F18 | 2011 | 0.0081 | 1.0310 | -0.0006 | 0.980 |
| F18 | 2012 | 0.5943 | 0.8498 | 0.0021 | 0.988 |
| F18 | 2013 | 0.5167 | 0.8549 | 0.0021 | 0.991 |

*Note*. Bold indicates the reference satellite in 2007.

The inter-calibration of $NDVI^{MVC}$ in the three overlaying years is shown in Fig. 4a (AVHRR) and Fig. 4b (MODIS). Via visual interpretation, the MODIS product has higher peak NDVI than AVHHR. The regression shows a linear relationship between the two $NDVI^{MVC}$ products ($R^2 = 0.934$) with $\alpha = 1.1835$ and $\beta = -0.1037$ (Fig. 4c). The histograms (Fig. 4d) demonstrate that the calibration process has shifted the AVHRR histogram to the right, making it more comparable with MODIS.

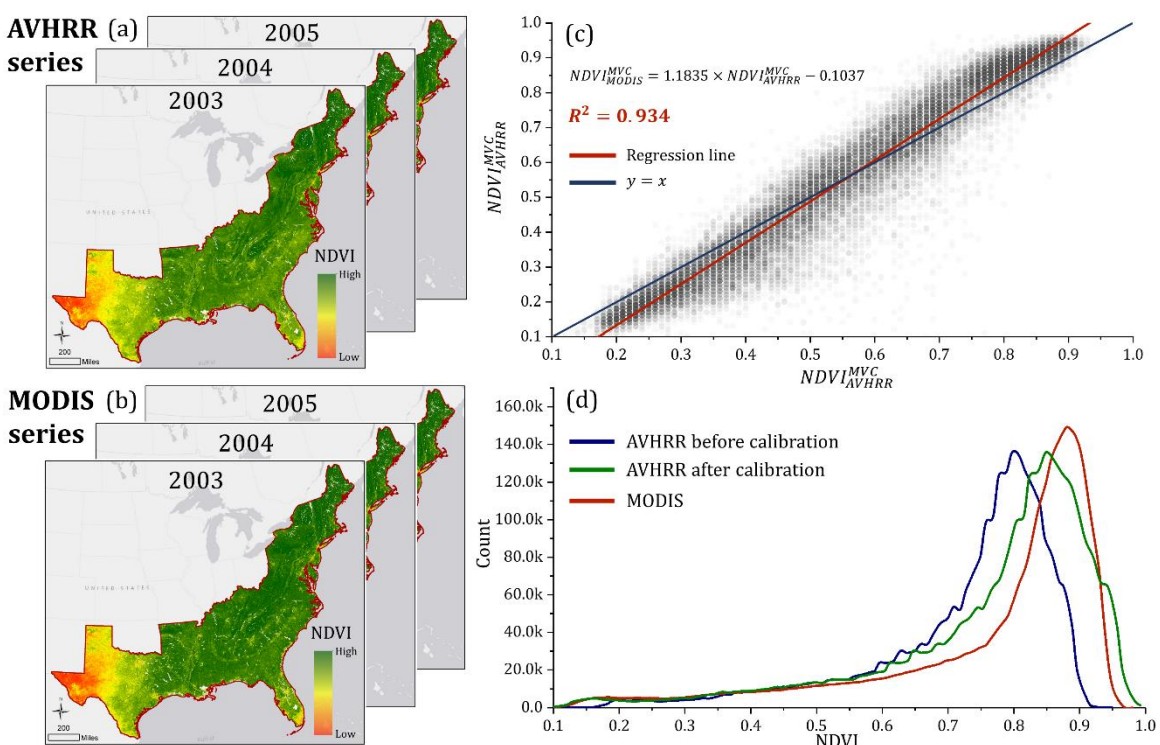

**Figure 4: (a)** $NDVI^{MVC}$ **series from AVHRR in the overlaying years; (b)** $NDVI^{MVC}$ **series from MODIS in the overlaying years; (c) Linear regression between AVHRR and MODIS using stratified sampling; (d) Comparison of histograms between MODIS and AVHRR (before and after calibration).**

## 5.3 The VANUI time series

An example VANUI map (1992) for the entire study area is shown in Fig. 5a, in which red color represents high VANUI value (high human settlement intensity) while blue color represents the opposite. Several subsets of the VANUI maps in years 1992, 2002 and 2013 are displayed to demonstrate more details in densely populated urban clusters: Philadelphia (Fig. 5b), Charlotte (Fig. 5c), Atlanta (Fig. 5d), Houston (Fig. 5e) and Orlando (Fig. 5f). Interestingly, the city of Philadelphia (Fig.5b) experienced slightly decreased human settlement intensity, especially in the 1992-2002 period. This observation agrees with the population dynamics of Philadelphia in the past decades: 1990-2000 (-4.3%), 2000-2010 (+0.6%). Similar trends of population decrease have been observed in other big northeastern cities such as Pittsburgh, in which its population dramatically decrease by -9.5% during 1990-2000 and -8.6% during 2000-2010 (U.S Census Bureau, 2018). The population loss is also recorded in a large number of small cities in the northeast region, including Johnstown and Rochester in NY, Weirton in WV and Harrisburg in PA (U.S Census Bureau, 2018).

Oppositely, the southern and southeastern cites have experienced intensified human settlement characterized by expanded city perimeters and intensified urban cores. Houston (Fig. 5e), for instance, has dramatically increased its human settlement. Again, this observation is well supported by the population boost per the census records, with an increasing rate of 19.8% in

1990-2000 and 7.5% in 2000-2010. Other cities, including Charlotte (Fig. 5c), Atlanta (Fig. 5d) and Orlando (Fig. 5f), also have seen significantly intensified human settlement supported by their increasing population records. In general, the opposite trends of human settlement between north and south of the study area match well with the "Snow Belt-to-Sun Belt" population shift trend documented in past studies in the last decades (Hogan, 1987; Iceland et al., 2013).

5    It could be noted that the VANUI maps in 2013 provide much finer details than those in 1992 and 2002. Given the unaltered spatial resolution of DMSP/OLS sensors, it can be explained by the different resolutions of the raw NDVI products from AVHRR (1km) and MODIS (250m). Although images have been resampled to the same pixel size (1km) and carefully calibrated in their time series, the intrinsic sensitivity of those two sensors still affects the VANUI outputs.

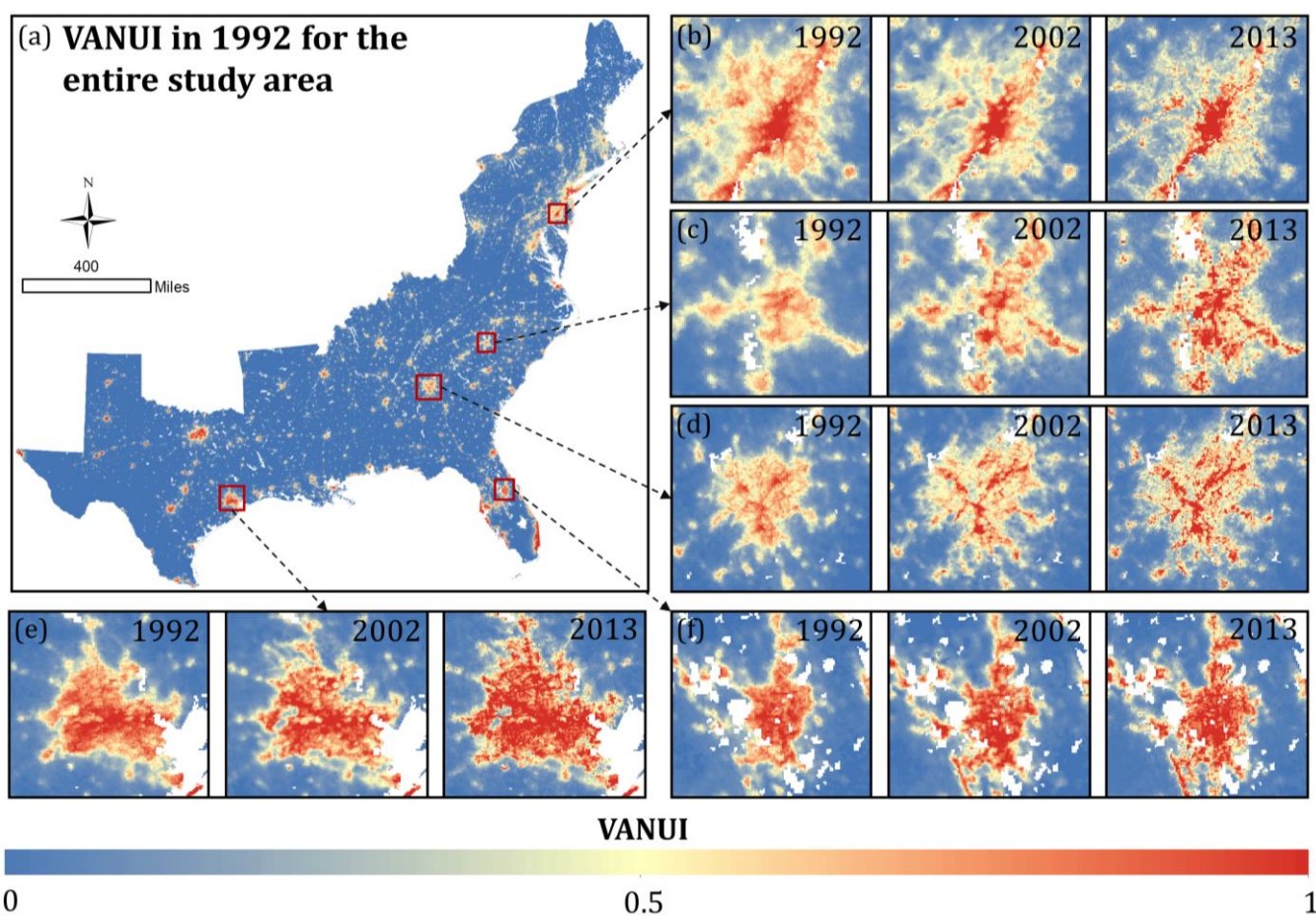

**Figure 5: The VANUI distribution in the study area in 1992 (a). The subfigures demonstrate the VANUI variations in 1992, 2002 and 2013 in five selected urban cities: Philadelphia (b), Charlotte (c), Atlanta (d), Houston (e) and Orlando (f). The white clusters are water bodies masked out of the analysis.**

## 5.4 Spatiotemporal patterns of human settlement and hurricane impacts

In each hurricane-prone zone, the yearly percentage lit pixels (VANUI> 0) sheds light on land development yearly, leading to a better understanding of the process of human settlement facing different degrees of hurricane impacts. The inter-annual fluctuation of total lit-pixel numbers exists in all zones, presumably due to the uncertainties introduced from the calibration of the DMSP/OLS NTL series and NDVI series. Bearing these noises, Fig. 6 presents the general trends of the lit pixel percentage in each zone. The lit pixel percentage varies in different zones, revealing a rank of Zone 1 (48.5%) followed by Zone 2 (45.4%), Zone 3 (41.6%) and Zone 4 (31.6%). Urban development was favored and prioritized in coastal regions, which were also the zones facing higher hurricane impacts.

As Fig. 6a (Zone 1) and Fig. 6b (Zone 2) suggest, the extent of human settlement in both zones increased significantly from 1992 to 2013, indicating consecutive land development in these highly hurricane-prone zones. The trends in both zones follow a logarithmic relationship that increased sharply in earlier years then slowed down. Located on the frontmost land-sea border, Zone 1 receives the most frequent and intense hurricane hits, yet its degree of fitness (coefficient of determination $R^2 = 0.898$) was higher than that of Zone 2 ($R^2 = 0.791$) in logarithmic regressions. With increased land development, we can conclude that the hurricane impacts on human settlement in these two zones are becoming more severe due to their higher hurricane-exposure. Zone 3 and Zone 4 are located further away from the coastal front. Although a slight increase lit pixel percentage could be visually observed for Zone 3 (Fig. 6c) and Zone 4 (Fig. 6d), their logarithmic trends are not statistically significant at confidence level $\alpha = 0.05$ and therefore, the regression lines are not marked in these figures. Fig.6 reveals a more significant increase in human settlement in areas closer to the coast front than inland during the 22-year period. The finding coincides with current literature in which studies reported the ever-growing population in coastal counties since the 1990s (Crosset, 2005; Stewart et al., 2003).

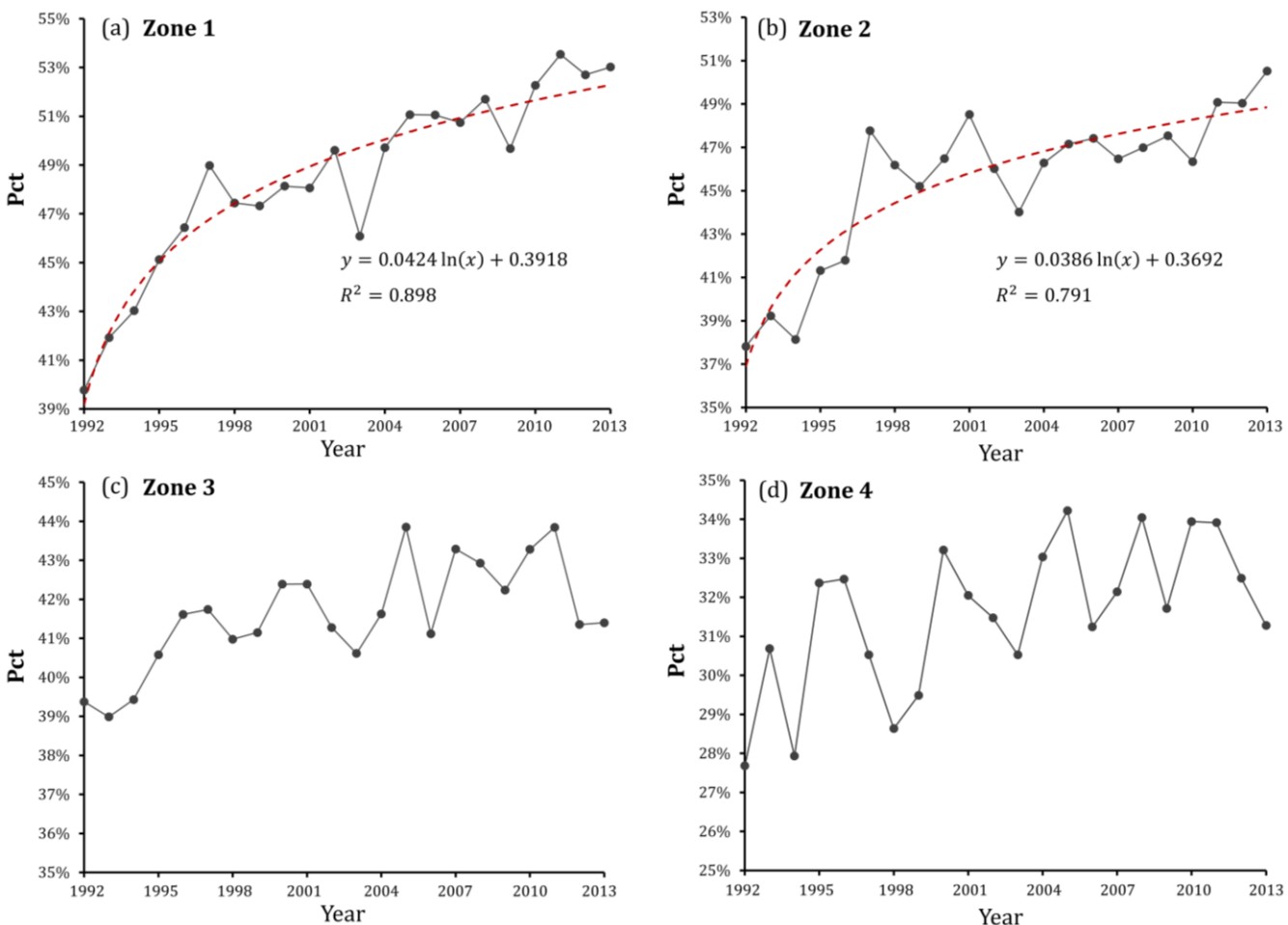

**Figure 6: Yearly statistics of percent area with VANUI larger than 0 in Zone 1 (a), Zone 2 (b), Zone 3 (c) and Zone 4 (d). In (a) and (b), the independent variable $x$ in the logarithmic regression model denotes the year sequence starting from 1992, meaning that $x = 1$ denotes the year 1992, $x = 2$ denotes 1993 and so on.**

The Mann-Kendall trend test coupled with Theil-Sen slope estimator extracted the areas with significant change (increase or decrease) of human settlement in the 22-year period (Fig. 7). Zonal statistics were also summarized for the four hurricane-prone zones (Table 3). The net increase area is defined as the area difference between pixels with a significant increasing and decreasing trend. The net increase zonal percentage represents the percentage of net increase area in each predefined hurricane-prone zone. As Table 3 suggests, 4.22% of the area in Zone 1 experienced a significant increase in human settlement, followed by 2.34% in Zone 2, 2.08% in Zone 3 and 1.65% in Zone 4. The statistics above suggest a noticeably positive relationship between the hurricane proneness of each zone and the percent area with a significant increase in settlement. The sum of Theil-Sen slope, on the other hand, established the relationship between hurricane proneness and the increase rate of settlement in each zone. Zone 1 receives the most hurricane hits but has the strongest increase of settlement intensity, followed by Zone 2, Zone 3, and Zone 4.

**Table 3.**
Hurricane-prone zonal summary of Mann-Kendall and Theil-Sen test

| Hurricane-prone zones | Zone size $(km^2)$ | Net increase area $(km^2)$[a] | Net increase zonal percentage (%) | Sum of Theil-Sen slope (per 100,000 $km^2$) |
|---|---|---|---|---|
| Zone 1 | 312,453 | 13,178 | 4.22 | 9.02 |
| Zone 2 | 507,285 | 11,889 | 2.34 | 6.11 |
| Zone 3 | 620,108 | 12,907 | 2.08 | 5.42 |
| Zone 4 | 1,047,424 | 17,255 | 1.65 | 4.16 |
| study area | 2,487,270 | 55,229 | 2.22 | 5.48 |

[a]Net increase area in each hurricane-prone zone denotes the area difference in this zone between pixels with significant increasing trend and pixels with a significant decreasing trend in their VANUI series.

Fig. 7a demonstrates the Mann-Kendall trend map in the study area where red, blue, and yellow in the figure represent pixel with a significant increasing trend, a significant decreasing trend, and an insignificant trend, respectively. Urban expansion of major cities in the south (the U.S. Southeast region), for example, Atlanta, Houston, and Dallas can be clearly observed as their city cores are surrounded by extensive areas with a significant increasing trend. A decrease in human settlement intensity was observed mostly in the north (the U.S. Northeast region; blue ellipse in Fig. 7a) where several cities in the state of New York stand out, including Albany, Troy, and Johnstown.

Two city clusters were selected to demonstrate the spatial distributions of the Mann-Kendall trend and Theil-Sen slope: Metro Atlanta, GA (Fig. 7b1-b2) and Metro Dallas, TX (Fig. 7c1-c2). For both cities, urban areas in 1992 were extracted from the Enhanced National Land Cover Data 1992 (NLCDe 92) released by U.S. Geological Survey (USGS) (https://water.usgs.gov/GIS/metadata/usgswrd/XML/nlcde92.xml), in which all classes including low intensity residential; high intensity residential; commercial/industrial/transportation and forest residential were counted as urban areas in 1992. From Fig.7, significant urban expansion can be observed in both cities. The growth of human settlement was also observed in small towns surrounding urban clusters.

For areas with a significant Mann-Kendall trend, the Theil-Sen slope indicates the change rate of human settlement (either upwards or downwards). In Fig. 7b2 and Fig. 7c2, the development of Metro Atlanta and Metro Dallas followed obvious radial patterns: areas close to the urban core showing a high increase rate of settlement (higher Theil-Sen slope) while areas away from urban core showing low increase rate. Since the VANUI has been normalized to [0,1] and the temporal period covers 22 years (1992-2013), a pixel would have a Theil-Sen slope of 0.045 (1/22), under the assumption that its settlement intensity had steadily increased from 0 in 1992 to 1 in 2013. The maximum Theil-Sen slope reached 0.037 in both cities, indicating a significant boost of human settlement intensity during the investigated period.

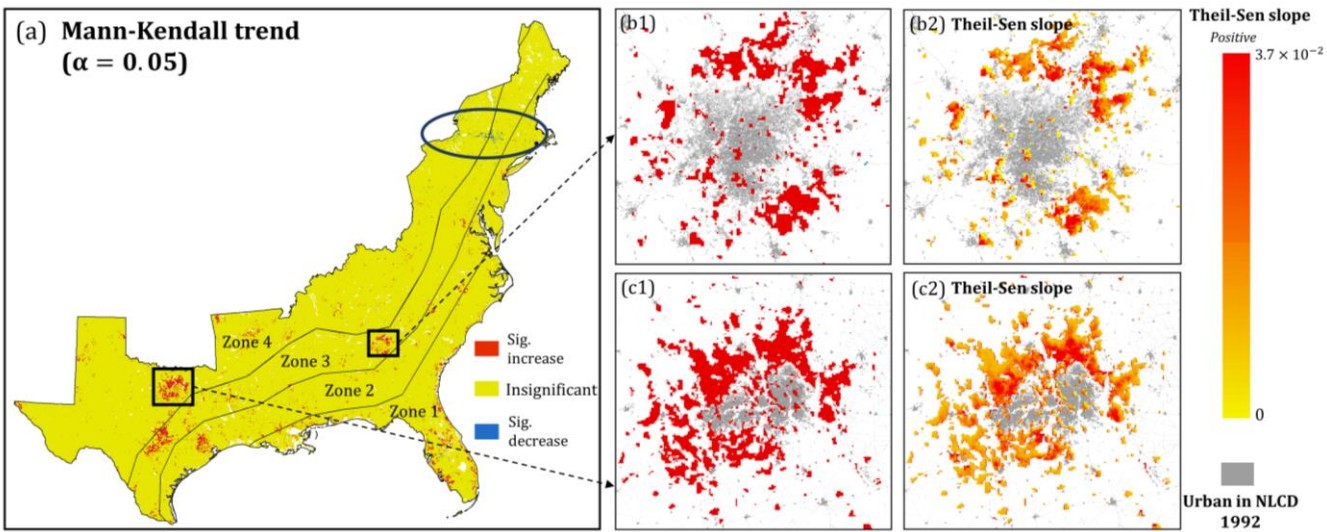

**Figure 7: Maps of the 22-year Mann-Kendall trend and Theil-Sen slope in the study area. Two subsets are selected: Dallas (trend map in b1 and slope map in b2) and Atlanta (trend map in c1 and slope map in c2).**

Metropolitan Statistical Areas (MSA) in the study area were selected for further analysis. Defined by the U.S Office of Management and Budget (OMB), MSA represents a contiguous area of relatively high population density. From a total of 383 predefined MSAs in the study area, the top 5 most populated MSAs in each part were selected. The lit pixel counts within the administrative boundary of each MSA in 1992, 2002 and 2013 were extracted. As Table 4 suggests, all selected MSAs in the north have decreased settlement intensities in two temporal periods (1992-2002 and 2002-2013). The only exception is the Washington-Arlington-Alexandria MSA in 2002-2013, during which its settlement intensity slightly increased by 2.5%. On the contrary, all of the top 5 most populated MSAs in the south witnessed a significant increase in settlement intensity. MSA of Dallas-Fort Worth-Arlington, for instance, has experienced a 23.8% increase of settlement intensity in 1992-2002 and the increase rate has slowed down to 4.6% in the next period (2002-2013). MSA of Miami-Fort Lauderdale-West Palm Beach, however, is believed to have a continuous boost of human settlement as its sum of VANUI has increased 12.6% in 1992-2002 and 11.3% in 2002-2013. Although four out of the five biggest MSAs in the south saw reduced growth rate in 2002 -2013 period (Table 4), Frey (2016) pointed that southern metropolitans have picked up their population increasing rate since 2015 and this could be a sign that southern metropolitans are heading back to the growth levels they experienced prior to the U.S recession in 2007 to 2009.

**Table 4.**

Sum of VANUI value and change percentage in the top 5 most populated MSA in the north and south of the study area

| MSAs[a] | Sum of VANUI in 1992 | Sum of VANUI in 2002 | Sum of VANUI in 2013 | % change (1992-2002) | % of change (2002-2013) |
|---|---|---|---|---|---|
| *North* | | | | | |
| New York-Newark-Jersey City | 3744.0 | 3307.2 | 3217.2 | -11.67% | -2.7% |

| | | | | | |
|---|---|---|---|---|---|
| Washington-Arlington-Alexandria | 1673.5 | 1611.4 | 1651.6 | -3.7% | +2.5% |
| Philadelphia-Camden-Wilmington | 2279.2 | 2068.1 | 1928.5 | -9.3% | -6.8% |
| Boston-Cambridge-Newton | 1498.9 | 1289.4 | 1182.3 | -14.0% | -8.3% |
| Baltimore-Columbia-Towson | 1035.5 | 961.2 | 831.2 | -7.2% | -13.5% |
| *South* | | | | | |
| Dallas-Fort Worth-Arlington | 3115.4 | 3857.1 | 4034.12 | +23.8% | +4.6% |
| Houston-The Woodlands-Sugar Land | 2687.0 | 3028.8 | 3143.9 | +12.7% | +3.8% |
| Miami-Fort Lauderdale-West Palm Beach | 1985.4 | 2262.7 | 2518.9 | +12.6% | +11.3% |
| Atlanta-Sandy Spring-Roswell | 2085.8 | 2398.8 | 2546.2 | +14.0% | +6.1% |
| Tampa-St. Petersburg-Clearwater | 1387.7 | 1511.9 | 1598.8 | +9.0% | +5.7% |

[a]All administrative boundaries of selected MSAs were derived from U.S Census Bureau: https://www.census.gov/geo/maps_data/data/cbf/cbf_msa.html. MSAs in the south were selected from Southeast and Gulf South of the U.S, and therefore, Washington-Arlington-Alexandria and Baltimore-Columbia-Towson were regarded as north MSAs in this study.

The ongoing intensification on human settlement in high hurricane-exposure areas, especially in the U.S southeastern region potentially leads to an escalation in flood-induced losses. Despite the fact that the driving factors are complex and unclear, they reflect the micro to macro levels of socioeconomic development that has been prioritized in high hurricane-exposure areas
in the last decades. Additionally, intensification of human settlement always couples with anthropogenic environmental changes (deforestation, wetland destruction, etc.), potentially resulting in more severe impacts during hurricanes and floods (Viero et al., 2019). Although the investigated period of this study stops at the year 2013 due to the termination of DMSP/OLS satellites, the intensification of human settlement in areas with high hurricane-exposure (like Zone 1) is expected to continue and might even accelerate. In alignment with economic recovery, studies have shown escalated population shift towards the
Atlantic and Gulf coast, after the stalling during the recession (Neumann et al., 2015).

Coastal resilience becomes more complicated when the increasing pressure of human settlement in coastal zones is coupled with the more frequent and costly hurricanes. The last three years (2016-2018) have seen consecutive above-average damaging Atlantic hurricane seasons. The economic damage in the conterminous U.S in 2017 was among the costliest ever recorded on a nominal, inflation-adjusted, and normalized basis (Klotzbach, 2018). What's worse, 2018 was the most recent hurricane
season to feature four simultaneously named storms (Florence, Isaac, Helene, and Joyce) after 2008. Although the future trend of hurricane seasons cannot be easily predicted, the implication of greater losses stands as the sizable growth of human settlement continues along the Atlantic and Gulf coasts.

With the launch of the Suomi National Polar-orbiting Partnership (NPP) Satellite in October 2011, NTL data from the Visible Infrared Imaging Radiometer Suite onboard have become available. Its on-board calibration capacity and saturation-free merit have made NPP-VIIRS a new generation system of nighttime light observations (Elvidge et al., 2013). This new NTL data source will provide improved monitoring of human settlement and land development in hurricane-prone regions for advanced disaster assessment.

## 6. Conclusion

This study examined the spatiotemporal dynamics of nighttime satellite-derived human settlement in 1992-2013 in four zones at different levels of hurricane proneness on the U.S. Atlantic and Gulf Coasts. The hurricane-prone zones were delineated based on historical storm tracks from the North Atlantic Basin during 1851-2016 via a wind speed weighted track density function. A three-step intercalibration framework was applied to intercalibrate the multi-satellite DMSP/OLS NTL series, and the NDVI-desaturated NTL products were extracted to derive VANUI, a popular index representing human settlement intensity. Mann-Kendall trend and Theil-Sen slope were further applied to identify the existing trend in the 22-year period.

Zonal statistics indicate that in the frontmost zones along the coast, i.e., Zone 1 and Zone 2 (receiving the most frequent hurricane hits), human settlement intensity has dramatically increased, although the change rate has slowed down since the early 2000s. The increase was not significant in areas farther away from the coasts (Zone 3 and Zone 4). Via trend analysis, 4.22% of the area in Zone 1 experienced a significant increase in settlement intensity, followed by 2.34% in Zone 2, 2.08% in Zone 3 and 1.65% in Zone 4, revealing higher pressure of human settlement and thus impacts from hurricanes in the frontmost coastal areas. Different from the zonal partitions, opposite trends of human settlement were observed from the north (decreasing) to the south (increasing) of the study region, which are supported by decadal census records. These opposite trends agree with the "Snow Belt-to-Sun Belt" U.S population shift reported in other studies. Along the Atlantic and Gulf coasts, the ongoing intensification of anthropogenic environmental changes coupled with more frequent and severe hurricanes is likely to cast more severe pressure on coastal resilience.

*Data availability.* All data used in this study are publicly available. The historical storm tracks were retrieved from International Best Track Archive for Climate Stewardship (IBTrACS), hosted by NOAA (https://www.ncdc.noaa.gov/ibtracs/). The DMSP/OLS NTL data were obtained from the National Centers for Environmental Information website (https://ngdc.noaa.gov/eog/dmsp/downloadV4composites.html). MODIS NDVI series was derived from Oak Ridge National Laboratory Distributed Active Archive Center (ORNL DAAC) (https://daac.ornl.gov/). AVHRR NDVI series was provided by the United States Geological Survey Earth Resources Observation and Science (USGS/EROS) (https://phenology.cr.usgs.gov/get_data_1km.php).

*Author contributions.* XH designed the analytical framework of this study, conducted GIS and statistical analysis, and drafted the manuscript. CW and JL provided methodological advice. XH, CW, and JL made major revisions of the manuscript.

*Competing interests.* The authors declare that they have no conflict of interest.

*Acknowledgements.* The authors would like to thank Huan Ning, the anonymous reviewers, and the editor for their constructive comments and suggestions, which greatly improved the quality of the manuscript.

*Financial support.* This study is partially supported by a 2018 ASPIRE Grant and a 2018 SPARC Graduate Research Grant
from the Office of the Vice Program, University of South Carolina.

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
