# Peer review of "Understanding Spatiotemporal Development of Human Settlement in Hurricane-prone Areas on U.S. Atlantic and Gulf Coasts using Nighttime Remote Sensing"

_Natural Hazards and Earth System Sciences, 2019_

## Referee Comment (RC1) · Anonymous Referee #1 · 5 Jun 2019

This manuscript describes the use of the DMSP/OLS NTL data from six satellites and AVHRR and MODIS optical imagery to derive the vegetation-adjusted NTL urban index (VANUI), which was analyzed using the Mann-Kendall test and Theil-Sen test for its spatiotemporal trend from 1992 to 2013. The VANUI product was then related to four hurricane-prone zones representing different levels of hurricane proneness, which was determined based on historical North Atlantic Basin (NAB)-origin storm tracks. The manuscript is well organized and the results are very encouraging. This manuscript is worth of being published, but the following comments would be helpful for the authors to

improve the quality of the manuscript. General comments: Line 12, page 2: it would be helpful to provide some background information on how a hurricane is categorized, and explain how a category 5 hurricane looks like. Line 13, page 2: Rephrase "125 billion and 50 billion dollars of damage respectively", this is confusing. Is the total damage 125 billion dollars? Or Is 125 billion dollars a part of damage? Current expression is more like the second case. The same clarification is needed for the 50 billion statement. Line 13, page 6: How is R determined? Based on what factors? Line 24, page 6: It would be helpful to make reference to Figure 3 when mentioning the referencing area. Line 28, page 6: how many referencing lit pixels are used? Figure 3b2, page 10: this plot is very scattered as compared to the other two plots, any explanations? Figure 4c, page 12: the level of the vertical axis is not correct. Figure 5, page 13: it is helpful to label which image is for 1992, 2002 and 2013. Figure 7a, page 17: the ellipse is shown without being explained.

Grammar errors can be found at some places though they don't prevent readers to understand the science of the paper. The following editorial comments could be helpful in this regards, yet it is unlikely that they are able to address all the language issues. Editorial comments: Line 28, page 2: a better understanding Line 34, page 2: Satellite to satellite Line 5, page 3: referred as to referred to as Line 16, page 3: were to was Line 18, page 3: significant to a significant Line 10, page 5: has to have Line 18, page 5: spell out USGS Line 27, page 6: in the same to at the same Line 10, page 7: year to years Line 25, page 7: on to on the Line 27, page 7: upwards or downwards to upward or downward trend Line 3, page 10: a R2 to an R2 Line 7, page 12: year to years Line 13, page 12: decreased to decrease by Line 9, page 12: City to the city Line 17, page 12: Houston reveals dramatic o Houston has a Line 3, page 13: matches to match Line 8, page 13: affect to affects Line 1, page 15: percentage of area to percent area Line 6, page 15: Zonal statistic to Zonal statistics; in to for Line 7, page 15: Please rephrase "The net increase area calculates" Line 11, page 15: significant to a significant Line 12, page 18: Please rephrase "The three two years (2016-2018)" Line 18, page 19: Rephrase "an areal percentage of 4.22% in Zone 1 experienced significant increase in

settlement intensity", it does not make sense.

---

## Referee Comment (RC2) · Anonymous Referee #2 · 6 Jun 2019

The paper presents an analysis of how and where human settlements increased across the USA in relation to hurricane locations using nighttime lights (NTL) from 1992 to 2013. The authors propose an alternative variable to NTL, VANUI – where NTL are linked to NDVI, to account for vegetation cover and also reduce the technical limitations of NTL data. Given the availability of time series, spatial and temporal dynamics are investigated, particularly focusing on hurricane-prone areas. The paper is relatively clearly written and I enjoyed reading it. However, before I recommend publication of this work, several clarifications need to be done. In particular, I do have one major

concern that may cause a major flaw in your research.

***To what extent the percentage of pixels with VANUI>0 is representative of an increased land development?*** If we look at the case of Philadelphia (Fig. 5b) the percentage does not significantly change from 1992 to 2013. What is changing here is the sum of VANUI values. Therefore, I strongly recommend to analyze the sum of VANUI to check for land development. If the sum of VANUI does not confirm your previous findings, then your analysis present a major flaw and cannot be accepted for publication.

***Why sum slope to represent the rapidness of human settlement growth?*** Each pixel (1 km2) can have a maximum slope value of 1/22=0.045. By summing slopes in a region, I assume that you can get an estimate of growth, but this is proportional to the considered area (namely, one of the four hurricane-prone zones), thus you cannot compare trends. To do this, you should consider the average slope.

***To what extent sum of slopes in Table 3 are significant?*** I do have serious concerns related to the slope values reported in Table 3. If we assume that the maximum slope per pixel is 0.045 and in zone 1 there are 312,453 pixels (all starting from 0 in 1992 and reaching 1 in 2013), then ideally the maximum sum value would be 14,060. How is this value related to 9.02 (per 100,000 km2)? Please elaborate more on this, to prove the significance of slope values.

Detailed comments Page 5 L. 13: "resampled to the 1 km pixel size". NTL are already at 1 km resolution.

Page 6 L. 20-21: categorization of hurricane-prone zones: what is the distribution of rho? You should consider (if not done yet) the frequency distribution of rho values to categorize zones.

L. 23-25: "Serving as the reference site in that study [Elvidge et al. 2009]. . .". Elvidge et al 2009 considered Sicily as the reference site to perform intercalibration, not Los

[Figure]

Angeles and the City of San Diego, as you stated in your manuiscript.

L. 30: Why do you calibrate DN values if you employ NDVI values? Zhang et al 2013 use original DN values. Please justify this.

Page 7 L. 10: 30,000 samples – are 30,000 samples per year or 10,000 samples per year? Is this representative of the range of NDVI values within the study area? If I am correct, you examine 0.4% (if 30k in total) or 1.2% (if 30k per year) of pixel to intercalibrate NDVI values.

Eq. 5: "k-1" should be "k=1". "j-k+1" should be "j=k+1"

Page 8 Eq. 6: "p-1" should be "p=1"

L. 9: add the meaning of Z=0 in this sentence

L. 24: Fig 2a instead of Fig 1a

Page 9 L. 1-4: Hew Hampshire is missing. Please add it to the list.

Page 14 L. 13-15: "With increased land development,…exposure." You cannot state this without examining the sum of VANUI.

Page 15 Caption of Fig 6: what does "and the like" mean here?

L. 6: I suggest to cite Fig 7 here

Page 17 L. 1-3: how is this linked to the sum of slopes in table 3?

Figure 7: You show in gray the urban area as derived from NLCD – does it show a yellow mann-kendall trend(i.e. insignificant)?

Page 18 L. 12: "The three two years": what does this mean?

Page 19 L. 15-17: You should tone down this sentence and verify if the sum of VANUI confirms your previous findings

References: Could you please list them in alphabetical order?

---

## Referee Comment (RC3) · Anonymous Referee #3 · 8 Jun 2019

The paper deals with an interesting analysis of a urbanization index change from 1992 to 2013 and storm proneness in the eastern coast of the United States, showing that the urbanization index is slightly decreased in area less prone to storms (north and far from the coast) and significantly increased in southern areas that are closer to the coast and far more prone to undergo severe storms. The topic is as interesting as important, and is dealt with by the Authors using modern data (remote sensing) and techniques. However, I have some major concerns that should be clarified and fixed before the paper can be accepted for publication.

[Figure]

**MAJOR POINTS**

-The goals of the study are not always declared clearly, and sometimes they are over-stated. For example, in the last paragraph of the introduction (p.3, l. 14) the Authors state that the goal of the paper is "to monitor urbanization process and hurricane im-pact". First, NTL is only a proxy for (some features of) urbanization, and "monitoring urbanization process" goes far beyond what is presented in this paper. Second, "hurri-cane impact" can be ascribed to a variety of factors (storm duration, exposure, vulnera-bility, etc.) that are not accounted for either by NTL or the wind speed only. To sum up, the paper draws a comparison between i) an urbanization index based on NTL (and not directly between urbanization) and ii) the storm proneness, unless the link between VANUI and real urbanization (or, better, exposure to storms) is validated quantitatively.

-I have some doubts on the significance of the linkage between satellite-derived in-dexes such as VANUI and urbanization of an area. First, the use of NTL-derive in-dexes as a proxy of urbanization intensities and exposure to storms should be vali-dated against urban maps and census data, at least for some significant regions/cities of the study area. Second, while it is evident that urbanization of rural areas produces an increased spatial extent of NTL, it is difficult for me to believe that a (moderate) de-crease of population in an already urbanized area would reflect in a reduction of NTL. To put it simply, the streetlights are not kept off because some apartments become un-inhabited, and the NTL differences linked to small population reductions are probably lower than uncertainties in the NTL data/calibration; buildings are rarely destroyed to restore cultivated fields. Rather, I see a very different resolution between 1992 and 2002 scenarios, which probably descends from the resolution of the NDVI. I think that substantial difference in the estimated extent of the urban area could descend from a sensibly different resolution of the processed data.

-Again about NTL. . . How can NTL be influenced by differences in how cities are illu-minated? For example, have policies been put in place to combat light pollution in the study area? (e.g., by forbidding upward oriented spots) How can these policies affect

the NTL?

-Page 8, l. 18-20: what does "per unit" mean? What is the sense of summing slopes?

-Note that urbanization process and exposure to storms can be monitored using census data. Such a way is undoubtedly more burdensome than using NTL, but far more precise and accurate.

SPECIFIC POINTS

-The paper contains a huge number of abbreviations, which sensibly hinder the text readability particularly for not-familiar readers. I ask the Authors to limit the number of abbreviations to the minimum necessary (for example, NAB, CONUS, EPB, DN are of course not necessary. . .).

-The quality of the English should be significantly improved.

-p.3, l. 6-9: applications of DMSP/OLS NTL data also encompass exposure to floods (Ceola et al., 2014, 2015).

-p.3, l. 20: what is "disaster migration"? Furthermore, an analysis of storm proneness can undoubtedly provide valuable information to support urban planning. The spatiotemporal changes of human settlement is what we need to influence, not an input data to allow disaster mitigation.

-Section 2 should be merged with the following sections into a "Material and methods" section.

-p.5, l. 3: "were downloaded". . . and also used? Or not??

-p.6, l. 13: "of radius R"

-p.6, l. 16: why using the wind speed and not the square of the speed? Consider that wind drag goes with the speed squared.

-p.19, l. 3: start a new paragraph after "the Atlantic Gulf coasts."

-I note that, in a recent study by Viero et al. (2019), similar trends have been identified (and conclusions drawn) for a large coastal lowland in Italy, where population has been found to resettle in areas at high(er) risk of flooding. Interesting comparisons could be drawn.

-In the bibliography, cited references should be ordered alphabetically. Please check all the bibliographic references throughout the text. For example, line 6 at page 2 should read "(Goldenberg et al., 2001)".

ADDITIONAL REFERENCES

Ceola, S., Laio, F., Montanari, A., 2014. Satellite nighttime lights reveal increasing human exposure to floods worldwide. Geophys. Res. Lett. 41, 7184–7190. doi:10.1002/2014GL061859

Ceola, S., Laio, F., Montanari, A., 2015. Human-impacted waters: New perspectives from global high-resolution monitoring. Water Resour. Res. 51, 7064–7079. doi:10.1002/2015WR017482

Viero, D.P., Roder, G., Matticchio, B., Defina, A., Tarolli, P., 2019. Floods, landscape modifications and population dynamics in anthropogenic coastal lowlands: The Polesine (northern Italy) case study. Sci. Total Environ. 651, 1435–1450. doi:10.1016/j.scitotenv.2018.09.121

---

## Author Comment (AC1) · 12 Jun 2019

**Response to first referee's comments on "Understanding Spatiotemporal Development of Human Settlement in Hurricane-prone Areas on U.S. Atlantic and Gulf Coasts using Nighttime Remote Sensing"**

By Xiao Huang, Cuizhen Wang and Junyu Lu

*"The manuscript is well organized and the results are very encouraging. This manuscript is worth of being published, but the following comments would be helpful for the authors to improve the quality of the manuscript."*

**Our response**: Thanks for your encouragement and your positive comments. Your suggestions are indeed helpful for us to improve the quality of this manuscript.

*"General comments: Line 12, page 2: it would be helpful to provide some background information on how a hurricane is categorized, and explain how a category 5 hurricane looks like"*

**Our response**: Thanks for your suggestions. We acknowledge that adding such information will benefit the readers. In the revision, we added some basic descriptions regarding the hurricane categorization: "Based on Saffir-Simpson Hurricane Scale, hurricane is categorized in five levels by its wind speed: 74-95 mph as Category 1; 96-110 mph as Category 2; 111-129 mph as Category 3; 130-156 mph as Category 4; above 157 mph as Category 5". In addition, we also stressed that category 5 is the highest category: "In 2016, Hurricane Mathew, a Category 5 (the highest category) hurricane, claimed a total of 34 direct deaths in U.S"

*"Line 13, page 2: Rephrase "125 billion and 50 billion dollars of damage respectively", this is confusing. Is the total damage 125 billion dollars? Or Is 125 billion dollars a part of damage? Current expression is more like the second case. The same clarification is needed for the 50 billion statement."*

**Our response**: Thanks for pointing out this issue. We acknowledge our description might cause confusion to the readers. We have revised the sentence as "In 2017, Hurricane Harvey in the Gulf coast caused a total of 125 billion dollars of damage, ranking the second costliest hurricanes in the U.S. In the same year, Hurricane Irma in the Atlantic coast caused a total of 50 billion dollars of damage, ranking the fifth costliest hurricanes in the U.S ("Costliest U.S. tropical cyclones tables updated", 2018)."

*"Line 13, page 6: How is R determined? Based on what factors?"*

**Our response**: We apologize that we didn't give enough information for the setting of circular neighborhood (R). For density calculations, a neighborhood size (or search distance) has to be

defined. We adopted the idea from the tool in ArcGIS called "Line Density" which calculates a magnitude-per-unit area from polyline features that fall within a radius around each cell. Here, we adopted the default setting of R in that function: "**The default is the shortest of the width or height of the output extent in the output spatial reference, divided by 30".** Per our calculation, the circular neighborhood R in our research area is 100 km. We added the setting of R in the revised manuscript as "The radius of R is set as 100 km in this study". We apologize for not specifying this parameter in the previous manuscript.

*"Line 24, page 6: It would be helpful to make reference to Figure 3 when mentioning the referencing area."*

**Our response**: Thanks for your suggestion. We added "Fig. 3a" to the sentence as "Serving as the reference site in that study **(Fig. 3a)**, the geographic area of metropolitan Los Angeles and City of San Diego, CA maintains high conformity of NTL values throughout the 22-year period (Kyba et al., 2017), which satisfies the "pseudo invariant" rule for calibration site selection (Elvidge et al., 2009)."

*"Line 28, page 6: how many referencing lit pixels are used?"*

**Our response**: Per our calculation, there are a total of 34,540 lit pixels (DN>0) in the reference site for our referencing satellite/year: F162007.

*"Figure 3b2, page 10: this plot is very scattered as compared to the other two plots, any explanations?"*

**Our response**: We believe that it is because F101992 is the very first satellite in the series. The long time interval from 2007 (our reference year) might lead to the scattered distribution pattern. However, we believe that the an $R^2$ of 0.946 still warrants a decent agreement for calibration. Our explanation in the manuscript is "The F101992 data (Fig. 3b2) exhibit less agreement due to its different satellite origin and a long time interval from 2007."

*"Figure 4c, page 12: the level of the vertical axis is not correct."*

**Our response**: We specifically select NDVI value above 0.1 to perform the intercalibration of MODIS and AVHRR. So, in the scatter plot, the origins should start from 0.1. In our previous manuscript, we illustrated: "A stratified sampling was applied to pixels with NDVI value above 0.1 to ensure that land covers in different NDVI ranges were equally sampled. A total of 30,000 samples were collected within four hurricane-prone zones in years 2003, 2004 and 2005."

*"Figure 5, page 13: it is helpful to label which image is for 1992, 2002 and 2013."*

**Our response**: Thanks for your suggestion. We labeled subfigures in Fig. 5. The new figure can be found here:

[Figure]

*"Figure 7a, page 17: the ellipse is shown without being explained."*

**Our response**: We explained the blue ellipse in our previous manuscript as "Decrease in human settlement intensity was observed mostly in the north (the U.S. Northeast region; **blue circle** in Fig. 7a) where several cities in state of New York stand out, including Albany, Troy and Johnstown." We changed "circle" to "ellipse" in this revision.

*"Grammar errors can be found at some places though they don't prevent readers to understand the science of the paper. The following editorial comments could be helpful in this regards, yet it is unlikely that they are able to address all the language issues."*

**Our response**: We really appreciate the grammar issue you pointed out in our manuscript. We have addressed all the grammar problems that have been pointed out. Before the submission of the revision, we will carefully proofread the document.

---

## Author Comment (AC2) · 16 Jun 2019

**Response to second referee's comments on "Understanding Spatiotemporal Development of Human Settlement in Hurricane-prone Areas on U.S. Atlantic and Gulf Coasts using Nighttime Remote Sensing"**

By Xiao Huang, Cuizhen Wang and Junyu Lu

*"The paper is relatively clearly written and I enjoyed reading it. However, before I recommend publication of this work, several clarifications need to be done. In particular, I do have one major concern that may cause a major flaw in your research"*

**Our response**: Thanks for your encouragement and comments. All your concerns were addressed in this response letter. Please find our point-by-point response below.

*"To what extent the percentage of pixels with VANUI>0 is representative of an increased land development? If we look at the case of Philadelphia (Fig. 5b) the percentage does not significantly change from 1992 to 2013. What is changing here is the sum of VANUI values. Therefore, I strongly recommend to analyze the sum of VANUI to check for land development. If the sum of VANUI does not confirm your previous findings, then your analysis presents a major flaw and cannot be accepted for publication."*

**Our response**: Thanks for pointing out the issue. We totally understand your concern. In the second round of our internal revision, we purposely changed "VANUI sum" to "percentage of pixels with VANUI > 0". Here are the seasons we'd like to present to you. Firstly, VANUI = (1-vegetation) * light. We believe that VANUI > 0 majorly represents lights in impervious surface area, not lights in vegetation area illuminated by cities ("1-vegetation index" solves the problem of light casted towards vegetation). For this reason, we believe that this yearly statistic illustrates the **expansion** of **newly** built area in different zones. Secondly, due to the saturation problem of DMSP (value capped at 63), we are worried that the sum of VANUI for **large-scale analysis** might suffer from some uncertainties (we did the sum of VANUI for different MSAs in CONUS, please see table 4). Thirdly, using the concept of "VANUI > a certain threshold" to perform regional analysis has been proved efficient by many (Li et al., 2016; Lu et al., 2018). In this study specifically, we set the threshold to 0 in order to capture all the newly expanded human settlement. To answer your concern, we attached figures after calculating the "Sum of VANUI". Please see attached figures for the obvious increase of sum of VANUI in Zone 1 and Zone 2 during the 90s:

[Figure]

Finally, after the investigation of the regional statistics, we did a Mann-Kendal test to investigate the trend for each single pixel during the 22-year period. In the trend analysis, we extracted all the pixels that exhibit significant increasing trend and summarized in each according zone. This statistic provides a better spatial explicit result than simply summing up VANUI in each zone. Our presenting logic follows:

1) Firstly, establish a general trend by investigating regional statistic summarized yearly (Figure 6)

2) Secondly, investigate trend of every single pixel via Mann-Kendall test, extract pixels with a significant increasing trend, and then summarize in each zone (Table 3)

*"Why sum slope to represent the rapidness of human settlement growth? Each pixel (1 km2) can have a maximum slope value of 1/22=0.045. By summing slopes in a region, I assume that you can get an estimate of growth, but this is proportional to the considered area (namely, one of the four hurricane-prone zones), thus you cannot compare trends. To do this, you should consider the average slope."*

**Our response**: Thanks for pointing the Theil-Sen slope issue. We totally agree with your opinion. Further, we'd like to explain our idea of using Theil-Sen slope. Mann-Kendall test only identifies pixels with significant trend (increasing or decreasing). It doesn't calculate the strength of the trend. After pixels are identified as having significant trend, Theil-Sen slope calculates how strong this trend is. By summing up all the Theil-Sen slopes in each zone, we can quantify the total increasing intensity in each zone. However, as you mentioned, this is proportional to the considered area. To solve this problem, we divided the summation of all the Theil-Sen slopes by the size of the zone. This is why (in Table 3) we added the column "Sum of Theil-Sen slope per 100,000 $km^2$". We are calculating the summation of Theil-Sen slope per unit in each zone.

*"To what extent sum of slopes in Table 3 are significant? I do have serious concerns related to the slope values reported in Table 3. If we assume that the maximum slope per pixel is 0.045 and in zone 1 there are 312,453 pixels (all starting from 0 in 1992 and reaching 1 in 2013), then ideally the maximum sum value would be 14,060. How is this value related to 9.02 (per 100,000 km2)? Please elaborate more on this, to prove the significance of slope values."*

**Our response:** Please let us explain our methodology (Mann-Kendall + Theil-Sen slope) in this study. Mann-Kendall identifies pixels with significant trend, and Theil-Sen further calculates the slope of those pixels that **have been identified**. As you pointed out, Zone 1 has 312,453 pixels. However, pixels with significant increasing trend only occupy 4.22% (Table 3), meaning that the we are only calculating the summation of Theil-Sen slope for around 13,185 pixels. Within those 13,185 pixels, only a small amount of pixels have increased from 0 (no urban at all) to 1 (fully urbanized). If, all the pixels in Zone 1 have max Theil-Sen slope (0.045), it should have a total of 593.3 (13,185 * 0.045). However, the total Theil-Sen slope in Zone 1 is around 28, meaning that pixels increasing from 0 to 1 occupy only a small percentage, which makes sense as areas that transform from pure rural (0) to pure urban (1) are limited in a developed country like U.S. In our perspective, the significance in Table 3 is not about the absolute value of "sum of Theil-Sen slope", but the comparison among different zones. After Mann-Kendall, as you may notice, the percentage of pixels with significant trend varies a lot in different zones (4.22% in Zone 1, nearly doubled from 2.34% in Zone 2). This statistic, however, only illustrates the coverage of significant pixels in different zones. It ignores the intensity of the increase. So we added the summation of Theil-

Sen slope in order to capture the "intensity" of this increase. As we expected, Zone 1 have not only the highest percentage of significant pixels but also the highest intensity of this increase. In our manuscript, we illustrated that "The net increase zonal percentage represents the percentage of net increase **area** in each predefined hurricane-prone zone…..The sum of Theil-Sen slope, on the other hand, established the relationship between the hurricane proneness and the **increase rate** of settlement in each zone". We appreciate your concerns and we will better elaborate this part in our revision.

*"Detailed comments Page 5 L. 13: "resampled to the 1 km pixel size". NTL are already at 1 km resolution."*

**Our response:** DMSP NTL series has a resolution of 30 arc second, which transforms to around 1 km at the equator. However, the pixel size of raw DMSP data varies a lot, given different latitudes. In this study, we resampled both NTL and NDVI into 1 km grid using Lambert Azimuthal Equal-area projection.

*"Page 6 L. 20-21: categorization of hurricane-prone zones: what is the distribution of rho? You should consider (if not done yet) the frequency distribution of rho values to categorize zones."*

**Our response:** Thanks for pointing out the categorization issue. We used Jenks optimization method (also called Jenks natural breaks classification method) to determine the arrangement of values into different class. The values in the sentence "Zone 4 (0-0.2), Zone 3 (0.2-0.5), Zone 2 (0.5-0.7) and Zone 1 (0.7-1.0)" are the rounded thresholds defined by Natural Jenks. We didn't include this information in the manuscript because we believe it is trivial compared to the entire workflow. Spearman's rho measures the strength of association between ranked variables. Since our density estimation is continuous, we believe Natural Jenks method is more valid in this case. We appreciate your understanding.

*"L. 23-25: "Serving as the reference site in that study [Elvidge et al. 2009]". Elvidge et al 2009 considered Sicily as the reference site to perform intercalibration, not Los Angeles and the City of San Diego, as you stated in your manuiscript."*

**Our response:** We sincerely apologize for the misuse of reference in our statement. Elvidge et al. (2009) did consider Sicily as reference site. In Hsu et al. (2015), they stated that "Los Angeles was taken as the reference for the Radiance Calibrated products for two reasons. First, Los Angeles has long been a mature metropolis and the light change is negligible. Second, being a metropolis, it can provide samples with high DNs from the city center, as well as low DNs from the suburban area". We have revised this sentence as "Serving as the reference site (Fig. 3a), the geographic area of metropolitan Los Angeles and City of San Diego, CA maintains high conformity of NTL values throughout the 22-year period (**Kyba et al., 2017; Hsu et al., 2015**), which satisfies the "pseudo invariant" rule for calibration site selection (Elvidge et al., 2009)"

*"L. 30: Why do you calibrate DN values if you employ NDVI values? Zhang et al 2013 use original DN values. Please justify this."*

**Our response:** In the study of Zhang et al. (2013), he proposed the calculation of VANUI but he did not focus on forming a time series. Since our NDVI were derived from two different satellites, we think it is valid to perform a calibration for two NDVI products as it is reported that NDVI for AVHRR and MODIS are different in a minor way (Gallo et al., 2004).

*"Page 7 L. 10: 30,000 samples – are 30,000 samples per year or 10,000 samples per year? Is this representative of the range of NDVI values within the study area? If I am correct, you examine 0.4% (if 30k in total) or 1.2% (if 30k per year) of pixel to intercalibrate NDVI values."*

**Our response:** Sorry for the confusion this sentence might cause. It is 30,000 samples per year. We have modified the sentence as "30,000 samples were collected within four hurricane-prone zones in years 2003, 2004 and 2005, respectively". In terms of the representativeness, we believe a total of 90,000 should be enough for a simple linear calibration. The result of $R^2 = 0.934$ in the calibration of AVHRR and MODIS proves that the products of those two satellites are very similar. Many studies have compared and calibrated NDVI, but their sample sizes are mainly within thousands (Beck, et al., 2011) or tens of thousands (Gallo et al., 2004).

*"Eq. 5: "k-1" should be "k=1". "j-k+1" should be "j=k+1""*

**Our response:** We appreciate your correction of our mistake. The mistakes in the function have been modified accordingly.

*"Page 8 Eq. 6: "p-1" should be "p=1""*

**Our response:** We appreciate your correction of our mistake. The mistake in the function has been modified accordingly.

*"L. 9: add the meaning of Z=0 in this sentence"*

**Our response:** Thank you for pointing out the missing explanation of Z = 0. We have revised the sentence as "The *Z* value in Eq.7 represents the monotonic tendency of a time series. A positive *Z* indicates an increasing trend while a negative *Z* indicates a decreasing one. **A stable trend exists when the value of Z equals 0**."

*"L. 24: Fig 2a instead of Fig 1a"*

**Our response:** We apologize for mislabeling of our figure. "Fig. 1a" has been replaced by "Fig. 2a"

*"Page 9 L. 1-4: Hew Hampshire is missing. Please add it to the list."*

**Our response:** We apologize for the missing New Hampshire in our list. This sentence has been revised as "The study area contains all U.S. states covered in the hurricane-prone zones (Fig. 2c): Maine, Massachusetts, New Jersey, New York, North Carolina, **New Hampshire**, Pennsylvania, Rhode Island, Tennessee, Texas, Maryland, Alabama, Arkansas, Connecticut, Delaware, DC, Florida, Georgia, Kentucky, Louisiana, Mississippi, South Carolina, Vermont, Virginia and West Virginia."

*"Page 14 L. 13-15: "With increased land development,: : :exposure." You cannot state this without examining the sum of VANUI."*

**Our response:** Please see our first response. We appreciate your understanding.

*"Page 15 Caption of Fig 6: what does "and the like" mean here?"*

**Our response:** We are sorry that the phrase "and the like" might cause confusion to our readers. The definition of "and the like" is more like the word "similarly". We do not think this phrase is a good fit in the sentence. We replaced "and the like" to "and so on" in this revised manuscript.

*"L. 6: I suggest to cite Fig 7 here"*

**Our response:** Thanks for your suggestion. We cited Fig 7 in the sentence you pointed out. The sentence has been modified as "The Mann-Kendall trend test coupled with Theil-Sen slope estimator extracted the areas with significant change (increase or decrease) of human settlement in the 22-year period **(Fig. 7)**"

*"Page 17 L. 1-3: how is this linked to the sum of slopes in table 3?"*

**Our response:** The Theil-Sen slope in our study case is relatively small (normalized VANUI value ranges from 0-1 and the time period investigated covers a 22-year period). The sentence you pointed out aims to explain value range of Theil-Sen slope in our study, providing an example of what Theil-Sen slope measures and how it measures.

*"Figure 7: You show in gray the urban area as derived from NLCD – does it show a yellow mann-kendall trend(i.e. insignificant)?"*

**Our response:** Our investigated time period starts from 1992 and this NLCD product shows the urban area in year 1992. Our idea is to provide a "base map", showing the urban extent in the very first year. The red pixels in Fig 7 (b1) and (c1) show all the areas identified as "significant increasing" during the 22-year period (1992-2013). Yes, some urban areas in 1992 show significant increasing trend as their urbanization might further intensify (Theil-Sen slope statistics explore this intensification of urbanization).

*"Page 18 L. 12: "The three two years": what does this mean?"*

**Our response:** We sincerely apologize for the typo here. The sentence has been revised as "The last two years (2016-2017) have seen a consecutive above-average damaging Atlantic hurricane season."

*"Page 19 L. 15-17: You should tone down this sentence and verify if the sum of VANUI confirms your previous findings"*

**Our response:** In terms of "sum of VANUI" issue, please see our first response. In this paragraph, we specifically illustrated the results from Mann-Kendall coupled with Theil-Sen slope. Whether a pixel exhibits significance in trends (increasing or decreasing) during the investigated period of time totally depends on the Mann-Kendall test. After the test, we summarized the number of pixels in each zone and did a zonal ratio. "Our result proves that 4.22% of the area in Zone 1 experienced significant increase in human settlement, followed by 2.34% in Zone 2, 2.08% in Zone 3 and 1.65% in Zone 4". This is the trend analysis result after we investigated the trend for every single pixel in our study area.

*"References: Could you please list them in alphabetical order?"*

**Our response:** Thanks for pointing out the issue. The references in the revised manuscript have been listed in alphabetical order.

**References used in this response:**

1. Beck, H. E., McVicar, T. R., van Dijk, A. I., Schellekens, J., de Jeu, R. A., & Bruijnzeel, L. A. (2011). Global evaluation of four AVHRR–NDVI data sets: Intercomparison and assessment against Landsat imagery. Remote Sensing of Environment, 115(10), 2547-2563.

2. Gallo, K., Ji, L., Reed, B., Dwyer, J., & Eidenshink, J. (2004). Comparison of MODIS and AVHRR 16-day normalized difference vegetation index composite data. Geophysical Research Letters, 31(7).

3. Hsu, F. C., Baugh, K., Ghosh, T., Zhizhin, M., & Elvidge, C. (2015). DMSP-OLS radiance calibrated nighttime lights time series with intercalibration. Remote Sensing, 7(2), 1855-1876.

4. Li, Q., Lu, L., Weng, Q., Xie, Y., & Guo, H. (2016). Monitoring urban dynamics in the southeast USA using time-series DMSP/OLS nightlight imagery. Remote Sensing, 8(7), 578.

5. Lu, H., Zhang, M., Sun, W., & Li, W. (2018). Expansion Analysis of Yangtze River Delta Urban Agglomeration Using DMSP/OLS Nighttime Light Imagery for 1993 to 2012. ISPRS International Journal of Geo-Information, 7(2), 52.

---

## Referee Comment (RC4) · Anonymous Referee #2 · 17 Jun 2019

I would like to thatnk the authors for their reply. However, I still have some doubts concerning the use of "percentage of pixels with VANUI>0" instead of "sum of VANUI". The authors, in their reply, state that "we believe that this yearly statistic [percentage of pixels] illustrates the expansion of newly built area in different zones".

I respectfully disagree. This yearly statistic considers all impervious surfaces, and does not take into account if it's a newly built area. Also, how can you identify a newly expanded human settlement only looking at percentage? TO do this, you should consider the sum of VANUI. Therefore, I kindly ask the authors to check, for the entire study area,

if the sum of VANUI confirms previous findings (namely, repeat exactly what you did with percentages).

I am totally fine with the trend analysis and the remaining replies.

---

## Author Comment (AC3) · 18 Jun 2019

**Response to third referee's comments on "Understanding Spatiotemporal Development of Human Settlement in Hurricane-prone Areas on U.S. Atlantic and Gulf Coasts using Nighttime Remote Sensing"**

By Xiao Huang, Cuizhen Wang and Junyu Lu

*"The topic is as interesting as important, and is dealt with by the Authors using modern data (remote sensing) and techniques. However, I have some major concerns that should be clarified and fixed before the paper can be accepted for publication."*

**Our response**: We appreciate your encouragement and comments. We will address all your concerns in the following response. Please find our point-by-point response below.

*"The goals of the study are not always declared clearly, and sometimes they are overstated. For example, in the last paragraph of the introduction (p.3, l. 14) the Authors state that the goal of the paper is "to monitor urbanization process and hurricane impact". First, NTL is only a proxy for (some features of) urbanization, and "monitoring urbanization process" goes far beyond what is presented in this paper. Second, "hurricane impact" can be ascribed to a variety of factors (storm duration, exposure, vulnerability, etc.) that are not accounted for either by NTL or the wind speed only. To sum up, the paper draws a comparison between i) an urbanization index based on NTL (and not directly between urbanization) and ii) the storm proneness, unless the link between VANUI and real urbanization (or, better, exposure to storms) is validated quantitatively."*

**Our response**: Thanks for your comments on the general goal of this study. We apologize for the overstatement in the last paragraph of the Introduction section. In this revision, we modified this paragraph to make it more suitable.

"The goal of this paper is to illustrate the use of DMSP/OLS NTL data in 1992-2013 to monitor urbanization process and hurricane impacts on the U.S. Atlantic and Gulf coasts **using nighttime artificial lights as proxy**…….. The spatiotemporal changes of human settlement **revealed from nighttime remote sensing** in **hurricane-prone zones** provide valuable information to evaluate damage and to support decision making of urban development."

We believe our modified version is more suitable given the context of this study. We appreciate your suggestion on our Introduction section.

*"I have some doubts on the significance of the linkage between satellite-derived indexes such as VANUI and urbanization of an area. First, the use of NTL-derive indexes as a proxy of urbanization intensities and exposure to storms should be validated against urban maps and census data, at least for some significant regions/cities of the study area."*

**Our response**: We totally understand your concerns regarding the satellite derived index and the exposure of storms. Please allow us to explain our insights on this one.

Firstly, we believe the focus of our article is not to prove/validate the linkage between satellite-derived indexes and urbanization. Rather, this study applies well established satellite-derived urban index. To make our approach convincing, we did a detailed review on the application of satellites, especially nighttime satellite (DMSP/OLS series). We illustrated "Extensive attempts have been made to harvest the NTL observations from DMSP/OLS in applications including urban expansion and decay (Lu et al., 2018), settlement dynamics (Elvidge et al., 1999; Yu et al., 2014), socioeconomic development (Doll et al., 2000) and energy consumption (Chand et al., 2009)". Further, in the newly added section 2 "Intercalibration and desaturation of DMSP/OLS NTL series", we did another review on the applications of nighttime satellite – derived activity index:

"A commonly used vegetation index, NDVI, is a useful indicator to reduce the saturation effect in DMSP/OLS data. Its practicality has been confirmed by many studies (Zhou et al., 2014; Liu et al., 2015). Lu et al. (2008) proposed a human settlement index (HSI) by merging normalized DMSP/OLS NTL data with the maximum NDVI in growing season derived from Moderate Resolution Imaging Spectroradiometer (MODIS). HSI has been proved rather efficient for settlement mapping in several testing sites in southeastern China. Zhang et al. (2013) develop a vegetation-adjusted NTL urban index (VANUI), which captures the inverse correlation between vegetation and luminosity. This simple index efficiently reveals the heterogeneity in regions with saturated DN values, which has been recognized by other study (Shao and Liu, 2014). Following the original design of NDVI that characterizes the inverse relationship between the near-infrared band and red band in vegetation, Zhang et al. (2015) designed a normalized difference urban index (NDUI) that characterizes the inverse relationship between vegetation and luminosity in a similar way. NDUI was evaluated in five testing sites in U.S and proved to be effective in desaturating DN values in DMSP/OLS."

We believe those aforementioned, widely recognized applications and the popularity of DMSP/OLS in long-term urban monitoring provide sufficient validity of our approach.

[Figure]

Fig. 9. A comparison of NTL based urban extents with high-resolution land cover products in China (top), US (middle), and Europe (bottom) at the cluster (left) and state/province/country (right) levels.

**(Comparison of NTL based urban extents with high-resolution land cover products by Zhou et al. 2018)**

As our study doesn't focus on proving the validity of satellite derived indices, we believe our approach has a great amount references as support for an application purpose. We sincerely appreciate your understanding.

In our manuscript, we did compare our observed trend (decrease of VANUI in the north) to the U.S Census Bureau. We illustrate in our manuscript:

"Similar trends of population decrease have been observed in other big northeastern cities such as Pittsburgh, in which its population dramatically decrease -9.5% during 1990-2000 and -8.6% during 2000-2010 (U.S Census Bureau, 2018). The population loss is also recorded in a large number of small cities in the northeast region including Johnstown and Rochester in NY, Weirton in WV and Harrisburg in PA (U.S Census Bureau, 2018)……In general, the opposite trends of human settlement between north and south of study area match well with the "Snow Belt-to-Sun Belt" population shift trend in the last decades that has been documented in past studies (Hogan, 1987; Iceland et al., 2013)."

In terms of quantifying storm exposure, we used the official storm tracks documented from 1851-2016. We didn't invent an exposure index in our study. Rather, we simply calculated the density

of those officially documented tracks. Our explanation on the "wind speed weighting" is presented in latter comments. We thank you for your valuable comments and suggestions.

**Additional references:**

Zhou, Y., Li, X., Asrar, G. R., Smith, S. J., & Imhoff, M. (2018). A global record of annual urban dynamics (1992–2013) from nighttime lights. *Remote Sensing of Environment*, *219*, 206-220.

*"Second, while it is evident that urbanization of rural areas produces an increased spatial extent of NTL, it is difficult for me to believe that a (moderate) decrease of population in an already urbanized area would reflect in a reduction of NTL. To put it simply, the streetlights are not kept off because some apartments become uninhabited, and the NTL differences linked to small population reductions are probably lower than uncertainties in the NTL data/calibration; buildings are rarely destroyed to restore cultivated fields. Rather, I see a very different resolution between 1992 and 2002 scenarios, which probably descends from the resolution of the NDVI. I think that substantial difference in the estimated extent of the urban area could descend from a sensibly different resolution of the processed data."*

**Our response:** Thanks for pointing out the decrease of NTL observed in our study. We well understand you concerns. In this response, we'd like to offer some evidence from other sources to back up the claim in our study. Firstly, we'd like to point out the existence of urban decay that has been stated in many studies:

1) "*Shrinking Cities in the United States of America*" by Pallagst (2009)
2) "*Viewing urban decay from the sky: a multi-scale analysis of residential vacancy in a shrinking U.S city*" by Deng and Ma (2015)
3) "*Shrinking Cities: Urban Challenges of Globalization*" by Martinez-Fernandez et al. (2012)
4) "*Ghost cities identification using multi-source remote sensing datasets: a case study in Yangtze River Delta*" by Zheng et al. (2017)
5) "*Ghost City Extraction and Rate Estimation in China Based on NPP-VIIRS Night-Time Light Data*" by Ge et al. (2018)

Some of those studies above documented the reduce of artificial lights in cities and they believe the decreasing of light in cities is partly due to the migration pattern and the suburbanization process. Secondly, to statistically investigate pixels that have a decreasing trend, we utilized Mann-Kendall test at a significant level of 0.05. We believe that trend test with this level of significance reduces the impact of the uncertainties in the calibration process and is able to extract pixels with significant decrease of VANUI value.

We acknowledge the difference of resolution between AVHRR (1 km) and MODIS (250 m). In this study, we resampled both of them to the same pixel size (1 km), carefully calibrated both of them in their 3 overlapping years using a total of 90,000 samples, and achieved an $R^2$ of 0.934. Based on the references that gave very promising NDVI calibration results between AVHRR and

MODIS (Tucker et al., 2005; Fensholt et al., 2009), and our very high $R^2$, we believe we have built a stable enough NDVI time series to be fused with our nighttime series. However, we do recognize the different sensitivity of those two sensors and we believe it inevitably leads to some uncertainties in our VANUI series. In our manuscript, we claim this potential uncertainty as:

"It could be noted that the VANUI maps in 2013 provide much finer details than those in 1992 and 2002. Given the unaltered spatial resolution of DMSP/OLS sensors, it can be explained by the different resolutions of the raw NDVI products from AVHRR (1km) and MODIS (250m). Although images have been resampled to the same pixel size (1km) and carefully calibrated in their time series, the intrinsic sensitivity of those two sensors still affect the VANUI outputs."

We thank you for your valuable comments.

**References used:**

Pallagst, K. (2009). Shrinking cities in the United States of America. *The Future of Shrinking Cities: Problems, Patterns and Strategies of Urban Transformation in a Global Context. Los Angeles (University of California)*, 81-88.

Deng, C., & Ma, J. (2015). Viewing urban decay from the sky: A multi-scale analysis of residential vacancy in a shrinking US city. *Landscape and Urban Planning*, *141*, 88-99.

Martinez-Fernandez, C., Audirac, I., Fol, S., & Cunningham-Sabot, E. (2012). Shrinking cities: Urban challenges of globalization. *International journal of urban and regional research*, *36*(2), 213-225.

Zheng, Q., Zeng, Y., Deng, J., Wang, K., Jiang, R., & Ye, Z. (2017). "Ghost cities" identification using multi-source remote sensing datasets: A case study in Yangtze River Delta. *Applied Geography*, *80*, 112-121.

Ge, W., Yang, H., Zhu, X., Ma, M., & Yang, Y. (2018). Ghost City Extraction and Rate Estimation in China Based on NPP-VIIRS Night-Time Light Data. *ISPRS International Journal of Geo-Information*, *7*(6), 219.

Frey, W. H. (2018). US population disperses to suburbs, exurbs, rural areas, and "middle of the country" metros,". *Brookings Institution*.

Kolko, J. (2017). Americans' Shift to the Suburbs Sped Up Last Year. *FiveThirtyEight*.

Tucker, C. J., Pinzon, J. E., Brown, M. E., Slayback, D. A., Pak, E. W., Mahoney, R., ... & El Saleous, N. (2005). An extended AVHRR 8-km NDVI dataset compatible with MODIS and SPOT vegetation NDVI data. *International Journal of Remote Sensing*, *26*(20), 4485-4498.

Fensholt, R., Rasmussen, K., Nielsen, T. T., & Mbow, C. (2009). Evaluation of earth observation based long term vegetation trends—Intercomparing NDVI time series trend analysis consistency of Sahel from AVHRR GIMMS, Terra MODIS and SPOT VGT data. *Remote Sensing of Environment*, *113*(9), 1886-1898.

*"Again about NTL. How can NTL be influenced by differences in how cities are illuminated? For example, have policies been put in place to combat light pollution in the study area? (e.g., by forbidding upward oriented spots) How can these policies affect the NTL?"*

**Our response:** Thanks for pointing out this issue. We acknowledge that there are many different ways in which cities are illuminated. In this study, however, we are not comparing one city to

another. We are comparing the same pixel (1 km by 1km) within its time series (1992-2013). We are aware that changes of policy regarding nighttime lights might affect the NTL. Since you pointed out the light pollution laws, we did a detailed research on all the policies regarding lights in U.S.

[Figure]

(Source: http://www.ncsl.org/research/environment-and-natural-resources/states-shut-out-light-pollution.aspx)

In our study area, TX, AR, FL, VA, DC, MD, DE, CT, RI, NY, NH, ME have established laws to combat light pollution. However, we noticed that during the investigated period (1992-2013), there is no significant change of policy for those states. As we have discussed in our manuscript, DMSP/OLS data is saturated at DN value of 63. Even if there is a sudden change of policy, we believe it won't significantly affect the urban core as it will still be capped as 63. We believe the popularity of DMSP/OLS in long-term urban studies in U.S has proved its effectiveness (Small et al., 2005; Zhang and Seto, 2011; Li et al., 2016).

**Additional references used:**

Small, C., Pozzi, F., & Elvidge, C. D. (2005). Spatial analysis of global urban extent from DMSP-OLS night lights. *Remote Sensing of Environment*, *96*(3-4), 277-291.

Zhang, Q., & Seto, K. C. (2011). Mapping urbanization dynamics at regional and global scales using multi-temporal DMSP/OLS nighttime light data. *Remote Sensing of Environment*, *115*(9), 2320-2329.

Li, Q., Lu, L., Weng, Q., Xie, Y., & Guo, H. (2016). Monitoring urban dynamics in the southeast USA using time-series DMSP/OLS nightlight imagery. *Remote Sensing*, *8*(7), 578.

*"Page 8, l. 18-20: what does "per unit" mean? What is the sense of summing slopes?"*

**Our response:** Please let us explain our methodology (Mann-Kendall + Theil-Sen slope) in this study. Mann-Kendall identifies pixels with significant trend, and Theil-Sen further calculates the slope of those pixels that have been identified. After the identification of pixels (1 km by 1 km area) with a significant increasing trend, we calculated the percentage of those pixels in each zone (Table 3). However, in our logic, this statistic only explores the "coverage" of areas with significant increase. Each pixel identified as significant has a "Theil-Sen" slope, representing how strong this increasing trend is. The sum of Theil-Sen slope calculates the entire increasing intensity in each zone. As you may notice, this summation is proportional to the size of different zones. To solve this, we simply normalized the summation of slope by dividing it by the size of each zone. This is why (in Table 3) we added the column "Sum of Theil-Sen slope per 100,000 $km2$". As we expected, Zone 1 have not only the highest percentage of significant pixels but also the highest intensity of this increase.

*"Note that urbanization process and exposure to storms can be monitored using census data. Such a way is undoubtedly more burdensome than using NTL, but far more precise and accurate."*

**Our response:** We'd like to provide our insight on this issue. Firstly, census data is based on samples, meaning that the quality of census data depends on the sample size and sampling period. As for American Community Survey (ACS), their 1-year estimates only survey areas with population larger than 65,000 and their 3-year estimates only survey areas with population larger than 20,000. The only product from ACS that covers all areas is their 5-year estimates, which is the estimation after 5 years of sampling. As for U.S Decennial Census, it provides surveyed results every 10 years based on sampling method. We believe nighttime remote sensing provides a better temporal explicit monitoring (yearly or even monthly) compared to traditional census data.

*(Source: https://www.census.gov/programs-surveys/acs/guidance/estimates.html)*

Secondly, census data tend to suffer from MAUP (Modified Areal Unit Problem) as the size of their unit varies a lot. Within each unit, however, we have to assume an uniform distribution of attributes from census data, which is not often the case. To some degree, nighttime remote sensing provides a better spatial explicit representation of human activity.

Here, we present blockgroups for a certain area in city of Atlanta to demonstrate the size difference of blockgroup as unit. We also present the distribution of buildings in blockgroups to demonstrate the great heterogeneity of buildings within this very small geographical level.

[Figure]

Blockgroups in City of Atlanta

Buildings within blockgroups

As you mentioned in your previous comments, in this study, NTL is just a proxy of monitoring human activity. We totally agree with your comments. We modified the last paragraph in our introduction section to better explain our goal.

"The goal of this paper is to illustrate the use of DMSP/OLS NTL data in 1992-2013 to monitor urbanization process and hurricane impacts on the U.S. Atlantic and Gulf coasts **using nighttime artificial lights as proxy**. Hurricane-prone areas were first derived by calculating the track density from historical storm tracks in the North Atlantic Basin. An intercalibrated DMSP/OLS NTL time series was built in a yearly interval. Assisted with the NDVI data, the Vegetation Adjusted NTL Urban Index (VANUI) was used to characterize human settlement intensities in the study area. After that, a trend analysis was conducted to identify areas with a significant increase of human settlement intensity in different zones. The spatiotemporal changes of human settlement **revealed from nighttime remote sensing** in hurricane-prone zones provide valuable information to evaluate damage and to support decision making of urban development."

*"The paper contains a huge number of abbreviations, which sensibly hinder the text readability particularly for not-familiar readers. I ask the Authors to limit the number of abbreviations to the minimum necessary (for example, NAB, CONUS, EPB, DN are of course not necessary."*

**Our response:** We agree that a large number of abbreviations might hinder the readability of our manuscript. Following your suggestion, we replaced "CONUS" to "the conterminous U.S", "NAB" to "North Atlantic Basin", "EPB" to "Eastern Pacific Basin". In terms of DN, we decided to keep the short form so that it can be consistent with the notation in our calibration functions. The aforementioned abbreviations in the figures and captions have been replaced as well:

[Figure]

**Figure 1: Historical storm tracks from the North Atlantic Basin (in red) and from the Eastern Pacific Basin (in green).**

*"The quality of the English should be significantly improved."*

**Our response:** Thanks for pointing out the language issue. In our revision, we performed spelling/grammatical check for the entire document. The revised manuscript has been carefully proofread multiple times and refined by native English speakers.

*"p.3, l. 6-9: applications of DMSP/OLS NTL data also encompass exposure to floods (Ceola et al., 2014, 2015)."*

**Our response:** Thanks for providing this relevant reference. We added those two references to our Introduction section as "In comparison, satellite-derived nighttime light (NTL) data provides a unique and direct observation of human settlement via night lights **(Ceola et al., 2014; Ceola et al., 2015**)."

**References used:**

1. Ceola, S., Laio, F., and Montanari, A.: Human-impacted waters: New perspectives from global high-resolution monitoring, Water Resour. Res., 51, 7064-7079, 2015.

2. Ceola, S., Laio, F., and Montanari, A.: Satellite nighttime lights reveal increasing human exposure to floods worldwide, Geophys. Res. Lett., 41, 7184-7190, 2014.

*"p.3, l. 20: what is "disaster migration"? Furthermore, an analysis of storm proneness can undoubtedly provide valuable information to support urban planning. The spatiotemporal changes of human settlement is what we need to influence, not an input data to allow disaster mitigation."*

**Our response:** We agree with your comment on disaster mitigation and we do not think disaster mitigation fits well in this context. This sentence has been revised as "The spatiotemporal changes of human settlement **revealed from nighttime remote sensing** in hurricane-prone zones provide valuable information to evaluate damage and to support decision making of urban development."

*"Section 2 should be merged with the following sections into a "Material and methods" section."*

**Our response:** Thanks for your suggestion on the organization of Section 2. We agree that some information in Section 2 can be merged to Section 3. In this revised manuscript, we reorganized the structure by merging information regarding the DMSP/OLS dataset to Section 3. We changed the title of Section 2 to "Intercalibration and desaturation of DMSP/OLS NTL series". In this new Section 2, we mainly focus on illustrating the limitations of DMSP/OLS series, explaining why we need to perform intercalibration and desaturation, and presenting the methods we choose to adopt. We believe a stand-alone section benefits the readers' understanding of this problem. This stand-alone section also helps to keep "Methods" section more focused and concise. In addition, we expanded section 2 to provide a better background of some famous efforts in addressing intercalibration and saturation of DMSP/OLS data. The newly expanded section is attached:

**2 Intercalibration and desaturation of DMSP/OLS NTL series**

Due to the absence of on-board calibration and intercalibration, the annual DMSP/OLS NTL composites derived from multiple satellites in a span of 22 years were not comparable directly (Li and Zhou, 2017; Liu et al., 2012). This lack of continuity and comparability has posed great challenges in DMSP/OLS NTL based trend analysis (Tan, 2016). Elvidge et al. (2009) designed a three-step framework to intercalibrate the DMSP/OLS NTL composites. Those three steps are: 1) selecting a reference region; 2) selecting a reference satellite year; 3) performing a 2nd-order polynomial regression against the NTL reference data. This simple framework has been proven efficient in reducing discrepancies in digital number (DN) values of the DMSP/OLS NTL time series (Pandey et al., 2013) and has been adopted in many studies (Liu and Leung, 2015; Huang et al., 2016).

Another notable limitation of DMSP/OLS NTL is the saturation of luminosity in the 6-bit (DN in a range of 0-63) imagery (Letu et al., 2010). Numerous attempts have been made to mitigate the saturation effect to retrieve the heterogeneity in areas with high intensity of human settlement. A commonly used vegetation index, NDVI, is a useful indicator to reduce the saturation effect in DMSP/OLS data. Its practicality has been confirmed by many studies (Zhou et al., 2014; Liu et al., 2015). Lu et al. (2008) proposed a human settlement index (HSI) by merging normalized DMSP/OLS NTL data with the maximum NDVI in growing season derived from Moderate Resolution Imaging Spectroradiometer (MODIS). HSI has been proved rather efficient for settlement mapping in several testing sites in southeastern China. Zhang et al. (2013) develop a vegetation-adjusted NTL urban index (VANUI), which captures the inverse correlation between vegetation and luminosity. This simple index efficiently reveals the heterogeneity in regions with saturated DN values, which has been recognized by other studies (Shao and Liu). Following the original design of NDVI that characterizes the inverse relationship between the near-infrared band and red band in vegetation, Zhang et al. (2015) designed a normalized difference urban index (NDUI) that characterizes the inverse relationship between vegetation and luminosity in a similar way. NDUI was evaluated in five testing sites in U.S and proved to be effective in desaturating DN values in DMSP/OLS.

In this study, the intercalibration of DMSP/OLS data follows the method proposed by Elvidge et al. (2009) and the desaturation of DMSP/OLS data is achieved by using VANUI (Zhang et al., 2013).

**Additional references added:**

1. Liu, Z., He, C., Zhang, Q., Huang, Q. and Yang, Y.: Extracting the dynamics of urban expansion in China using DMSP-OLS nighttime light data from 1992 to 2008, Landscape Urban Plann., 106, 62-72, 2012.

2. Li, X., and Zhou, Y.: Urban mapping using DMSP/OLS stable night-time light: a review. Int. J. Remote Sens., 38, 6030-6046, 2017.

3. Shao, Z. and Liu, C.: The integrated use of DMSP-OLS nighttime light and MODIS data for monitoring large-scale impervious surface dynamics: A case study in the Yangtze River Delta. Remote Sens., 6(10), 9359-9378, 2014.

4. Zhang, Q., Li, B., Thau, D., and Moore, R.: Building a better urban picture: Combining day and night remote sensing imagery, Remote Sens., 7, 11887-11913, 2015.

*"p.5, l. 3: "were downloaded": : : and also used? Or not??"*

**Our response:** We replaced word "downloaded" with word "used" in this revision.

*"of radius R"*

**Our response:** We apologize that we didn't give enough information for the setting of circular neighborhood (R). In this case, R represents a **circular domain** and its radius (one of its attributes) defines the size of this domain. We adopted the idea from the tool in ArcGIS called "Line Density" which calculates a magnitude-per-unit area from polyline features that fall within a radius around each cell. Here, we adopted the default setting of radius of R in that function: "The default is the shortest of the width or height of the output extent in the output spatial reference, divided by 30". Per our calculation, the circular neighborhood R in our research area is 100 km. We added the setting of R in the revised manuscript as "The radius of R is set as 100 km in this study". We apologize for not specifying this parameter in the previous manuscript.

*"why using the wind speed and not the square of the speed? Consider that wind drag goes with the speed squared."*

**Our response:** The reason of introducing "wind speed weighted track density" in our study is to generally categorize the severity of hurricane exposure. We understand that our weighting scheme may not be the only option. It is our assumption. The relationship between wind speed and damage is never fixed and we acknowledge that it depends on many factors. Besides, damage resulting from wind only consists part of the total damage introduced from hurricanes. Since wind speed is the best attribute in the dataset to distinguish different levels of the storm, and given the generally positive relationship between wind speed and hurricane impact, we assume a simple linear function in this study. In some studies, the relationship between wind speed and damage is conceptualized as:

[Figure]

("Dealing to wind hazards in New Zealand" from
https://www.niwa.co.nz/sites/niwa.co.nz/files/import/attachments/wind2.pdf)

[Figure]

Fig. 11. Wind damage band for 1–3 story residential buildings. ♦ Upper, ■ lower.

(Unanwa et al., 2000)

From those figures, we can observe that for three most common hurricane types (Cat 2, 3, and 4), their wind speed and damage relations can be assumed linear. We appreciate your understanding.

*"p.19, l. 3: start a new paragraph after "the Atlantic Gulf coasts."*

**Our response:** A new paragraph was started in the revision. We appreciate your suggestion.

*"I note that, in a recent study by Viero et al. (2019), similar trends have been identified (and conclusions drawn) for a large coastal lowland in Italy, where population has been found to resettle in areas at high(er) risk of flooding. Interesting comparisons could be drawn."*

**Our response:** Thanks for providing this reference. We acknowledge the close relationship between this reference and our study. In their article, they pointed out that anthropogenic landscape modifications can significantly affect flood hazard. We found this statement extremely helpful in backing up one of our statements. In this revision, we added this reference in the following context: "Additionally, intensification of human settlement always couples with anthropogenic environmental changes (deforestation, wetland destruction, etc.), potentially resulting in more severe impacts during hurricanes and floods (Viero et al., 2019)."

*"In the bibliography, cited references should be ordered alphabetically. Please check all the bibliographic references throughout the text. For example, line 6 at page 2 should read "(Goldenberg et al., 2001)".*

**Our response:** We apologize for the mistake of the intext citation you pointed out. We have checked all the references and the references in the revised manuscript have been listed in alphabetical order.

*"ADDITIONAL REFERENCES"*

**Our response:** All the additional references have been added per your suggestion. The references added in the revision include:

1. Ceola, S., Laio, F., & Montanari, A. (2014). Satellite nighttime lights reveal increasing human exposure to floods worldwide. Geophysical Research Letters, 41(20), 7184-7190.

2. Ceola, S., Laio, F., & Montanari, A. (2015). Human-impacted waters: New perspectives from global high-resolution monitoring. Water Resources Research, 51(9), 7064-7079.

3. Viero, D. P., Roder, G., Matticchio, B., Defina, A., & Tarolli, P. (2019). Floods, landscape modifications and population dynamics in anthropogenic coastal lowlands: The Polesine (northern Italy) case study. Science of The Total Environment, 651, 1435-1450.

**Other references used in this response:**

Unanwa, C. O., McDonald, J. R., Mehta, K. C., & Smith, D. A. (2000). The development of wind damage bands for buildings. *Journal of Wind Engineering and Industrial Aerodynamics*, *84*(1), 119-149.

"Dealing to wind hazards in New Zealand" from https://www.niwa.co.nz/sites/niwa.co.nz/files/import/attachments/wind2.pdf

---

## Author Comment (AC4) · 18 Jun 2019

**Response to second referee's comments on "Understanding Spatiotemporal Development of Human Settlement in Hurricane-prone Areas on U.S. Atlantic and Gulf Coasts using Nighttime Remote Sensing" (round 2)**

By Xiao Huang, Cuizhen Wang and Junyu Lu

*"I would like to thank the authors for their reply. However, I still have some doubts concerning the use of "percentage of pixels with VANUI>0" instead of "sum of VANUI". The authors, in their reply, state that "we believe that this yearly statistic [percentage of pixels] illustrates the expansion of newly built area in different zones"."*

**Our response**: Thank you for your comment. We well understand your concern. Please allow us to correct our claim "*we believe that this yearly statistic [percentage of pixels] illustrates the expansion of newly built area in different zones*". We meant to say that the **evolvement** of percentage of pixels (VANUI > 0) **during the 22-year investigated period** illustrates the trend of the expansion in each zone. If in Zone 1, there are 30% of pixels (VANUI > 0) in 1992 and 10 year later, there are 40% of pixels (VANUI > 0) in 2002, we believe the difference in the percentage illustrates the expansion of human settlement. In our previous manuscript, we aimed to build the percentage of pixels with VANUI > 0 at a yearly basis (Figure 6), and we did find remarkable trend in both Zone 1 and Zone 2, following a log distribution.

*"I respectfully disagree. This yearly statistic considers all impervious surfaces, and does not take into account if it's a newly built area. Also, how can you identify a newly expanded human settlement only looking at percentage?"*

**Our response**: Thanks for providing your insight on this issue. We respectfully disagree with the claim that VANUI considers all impervious surfaces. Actually, based on the calculation of VANUI:

VANUI = light * (1-vegetation)

we could claim that VANUI > 0 represents "**lights AND impervious surface",** meaning that it has to satisfy two requirements: 1) lights casted to a certain area and 2) that area has to have impervious surface. Traditional remote sensing (multispectral) that gauges urban expansion mainly consider the expansion of impervious surface. VANUI transcends those approaches by introducing another aspect of "lights". By building a time series (yearly) of percentage of pixels (VANUI > 0), we believe we can gauge the urban expansion trend in each zone. We'd like to apologize for our claim "identify newly expanded human settlement" in our response letter. We believe Figure 6 only presents the evolvement of percentage during the 22-year period. The

identification of newly expanded human settlement is completed by using our trend analysis approach (Mann-Kendall + Theil-Sen slope).

*"To do this, you should consider the sum of VANUI. Therefore, I kindly ask the authors to check, for the entire study area, if the sum of VANUI confirms previous findings (namely, repeat exactly what you did with percentages)."*

**Our response**: Thank you for your suggestion. We confirmed that the trend of VANUI summation in different zones agrees well with what we have found using percentage of VANUI>0. We performed the statistics for Zone 1 and Zone 2 using sum of VANUI and found the same log trend while Zone 3 and Zone 4 exhibit no significant trend. We presented the yearly sum of VANUI in Zone 1 and Zone 2 in the previous response.

*"I am totally fine with the trend analysis and the remaining replies."*

**Our response**: We are glad that you are satisfied with our response. We appreciate your valuable comments. Thanks for your time.

---

## Referee Comment (RC5) · Anonymous Referee #2 · 19 Jun 2019

I am fine with the replies and I look forward the publication of the manuscript.
* * *

---

## Author Comment (AC5) · 26 Jun 2019

We are glad that you are satisfied with our responses. We sincerely appreciate your time.

---

## Author Response (AR1)

**Response to RC 1**

Dear first referee,

Thank you for providing such valuable comments and suggestions during the open discussion process. We provided detailed point-by-point responses to your concerns during our discussion and here, we'd like to reiterate our responses and provide details about our revision in a table.

| Comments                                                                                                                                                                                                                                                                                                                  | Responses                                                                                                                                                                                                                                                                                                                                                                                                                                                                                                                                                                                                                                      |
|---------------------------------------------------------------------------------------------------------------------------------------------------------------------------------------------------------------------------------------------------------------------------------------------------------------------------|------------------------------------------------------------------------------------------------------------------------------------------------------------------------------------------------------------------------------------------------------------------------------------------------------------------------------------------------------------------------------------------------------------------------------------------------------------------------------------------------------------------------------------------------------------------------------------------------------------------------------------------------|
| "The manuscript is well organized and the results are very encouraging. This manuscript is worth of being published, but the following comments would be helpful for the authors to improve the quality of the manuscript."                                                                                               | Thanks for your encouragement and your positive comments. Your suggestions are indeed helpful for us to improve the quality of this manuscript.                                                                                                                                                                                                                                                                                                                                                                                                                                                                                                |
| Line 12, page 2: it would be helpful to provide some background information on
how a hurricane is categorized, and explain how a category 5 hurricane looks
like"                                                                                                                                                   | Thanks for your suggestions. We acknowledge that adding such information will benefit the readers. In the revision, we added some basic descriptions regarding the hurricane categorization: "Based on Saffir-Simpson Hurricane Scale, a hurricane is categorized in five levels by its wind speed: 74-95 mph as Category 1; 96-110 mph as Category 2; 111-129 mph as Category 3; 130-156 mph as Category 4; above 157 mph as Category 5.". In addition, we also stressed that category 5 is the highest category: "In 2016, Hurricane Mathew, a Category 5 (the highest category) hurricane, claimed a total of 34 direct deaths in U.S."     |
|                                                                                                                                                                                                                                                                                                                           | See Page 2, Line 4-6 and line 14-15                                                                                                                                                                                                                                                                                                                                                                                                                                                                                                                                                                                                            |
| "Line 13, page 2: Rephrase "125 billion and 50 billion dollars of damage
respectively", this is confusing. Is the total damage 125 billion dollars? Or Is
125 billion dollars a part of damage? Current expression is more like the
second case. The same clarification is needed for the 50 billion statement." | Thanks for pointing out this issue. We acknowledge our description might
cause confusion to the readers. We have revised the sentence as "In 2017,
Hurricane Harvey in the Gulf coast caused a total of 125 billion dollars of
damage, ranking the second costliest hurricanes in the U.S. In the same year,
Hurricane Irma in the Atlantic coast caused a total of 50 billion dollars of
damage, ranking the fifth costliest hurricanes in the U.S ("Costliest U.S.
tropical cyclones tables updated", 2018)."                                                                                                              |
|                                                                                                                                                                                                                                                                                                                           | See Page 2, Line 13-18                                                                                                                                                                                                                                                                                                                                                                                                                                                                                                                                                                                                                         |
| "Line 13, page 6: How is R determined? Based on what factors?"                                                                                                                                                                                                                                                            | We apologize that we didn't give enough information for the setting of circular neighborhood (R). For density calculations, a neighborhood size (or search distance) has to be defined. We adopted the idea from the tool in ArcGIS called "Line Density" which calculates a magnitude-per-unit area from polyline features that fall within a radius around each cell. Here, we adopted the default setting of R in that function: "The default is the shortest of the width or height of the output extent in the output spatial reference, divided by 30". Per our calculation, the circular neighborhood R in our research area is 100 km. |

|                                                                                                                   | We added the setting of R in the revised manuscript as "The radius of R is set
as 100 km in this study". We apologize for not specifying this parameter in the
previous manuscript.                                                                                                                                                                                                                                                                                                                               |
|-------------------------------------------------------------------------------------------------------------------|-------------------------------------------------------------------------------------------------------------------------------------------------------------------------------------------------------------------------------------------------------------------------------------------------------------------------------------------------------------------------------------------------------------------------------------------------------------------------------------------------------------------------|
| "Line 24, page 6: It would be helpful to make reference to Figure 3 when mentioning the referencing area." | Thanks for your suggestion. We added "Fig. 3a" to the sentence as "We adopted
the Elvidge et al. (2009) procedure to intercalibrate the DMSP/OLS NTL time
series. Serving as the reference site (Fig. 3a), the geographic area of
metropolitan Los Angeles and City of San Diego, CA maintains high
conformity of NTL values throughout the 22-year period (Kyba et al., 2017;
Hsu et al., 2015), which satisfies the "pseudo invariant" rule for calibration site
selection (Elvidge et al., 2009)." |
|                                                                                                                   | See Page 7, Line 15-18                                                                                                                                                                                                                                                                                                                                                                                                                                                                                                  |
| "Line 28, page 6: how many referencing lit pixels are used?"                                                      | Per our calculation, there are a total of 34,540 lit pixels (DN>0) in the reference site for our referencing satellite/year: F162007.                                                                                                                                                                                                                                                                                                                                                                                   |
| "Figure 3b2, page 10: this plot is very scattered as compared to the other two plots, any explanations?"          | We believe that it is because F101992 is the very first satellite in the series.
The long time interval from 2007 (our reference year) might lead to the
scattered distribution pattern. However, we believe that the an $R^2$ of 0.946
still warrants a decent agreement for calibration. Our explanation in the
manuscript is "The F101992 data (Fig. 3b2) exhibit less agreement due to its
different satellite origin and a long time interval from 2007."                                           |
|                                                                                                                   | See Page 10, Line 10-11                                                                                                                                                                                                                                                                                                                                                                                                                                                                                                 |
| "Figure 4c, page 12: the level of the vertical axis is not correct."                                              | We specifically select NDVI value above 0.1 to perform the intercalibration of MODIS and AVHRR. So, in the scatter plot, the origins should start from 0.1. In our previous manuscript, we illustrated: "Stratified sampling was applied to pixels with NDVI value above 0.1 to ensure that land covers in different NDVI ranges were equally sampled. Thirty thousand samples were collected within four hurricane-prone zones in years 2003, 2004 and 2005, respectively."                                            |
|                                                                                                                   | See Page 7, Line 31, and Page 8, Line 1-2                                                                                                                                                                                                                                                                                                                                                                                                                                                                               |

| "Figure 5, page 13: it is helpful to label which image is for 1992, 2002 and 2013." | Thanks for your suggestion. We labeled subfigures in Fig. 5. The new figure can be found here:
(a) VANUI in 1992 for the entire study area (a) (b) (c) (c) (c) (c) (c) (c) (c) (c) (c) (c                                                                                                                                                                                                     |
|-------------------------------------------------------------------------------------|--------------------------------------------------------------------------------------------------------------------------------------------------------------------------------------------------------------------------------------------------------------------------------------------------------------------------------------------------------------------------------------------------|
| "Figure 7a, page 17, the ellipse is shown without hoing evaluated"                  | See Page 14, Figure 5                                                                                                                                                                                                                                                                                                                                                                            |
| Figure 74, page 17. the ettpse is shown without being explained.                    |  <li>we explained the onde empse in our previous manuscript as "A decrease in human settlement intensity was observed mostly in the north (the U.S. Northeast region; blue ellipse in Fig. 7a) where several cities in the state of New York stand out, including Albany, Troy, and Johnstown." We changed "circle" to "ellipse" in this revision.</li> <li>See Page 17, Line 6-8</li>  |
| "Line 28, page 2: a better understanding"                                           | We modified the sentence as: "a better understanding of the temporal and
spatial dynamics of human settlement is needed for advanced damage
assessment and sustainable urban planning."                                                                                                                                                                                                    |
|                                                                                     | See Page 2, Line 30-31                                                                                                                                                                                                                                                                                                                                                                           |
| "Line 34, page 2: Satellite to satellite"                                           | "Satellite-derived" was modified to "satellite-derived"                                                                                                                                                                                                                                                                                                                                          |
|                                                                                     |                                                                                                                                                                                                                                                                                                                                                                                                  |
| "Line 5, page 3: referred as to referred to as"                                     | "reterred as" was modified to "reterred to as"                                                                                                                                                                                                                                                                                                                                                   |

|                                                                     | See Page 3, Line 10                                                       |
|---------------------------------------------------------------------|---------------------------------------------------------------------------|
| "Line 16, page 3: were to was"                                      | "were" was modified to "was"                                              |
|                                                                     | See Page 3 Line 22                                                        |
| "Line 18, page 3: significant to a significant"                     | "significant" was modified to "a significant"                             |
|                                                                     |                                                                           |
|                                                                     | See Page 3, Line 24                                                       |
| "Line 10, page 5: has to have"                                      | "has" was modified to "have".                                             |
|                                                                     | See Page 6, Line 5                                                        |
| Line 18, page 5: spell out USGS                                     | The sentence has been modified as "provided by United States Geological   |
|                                                                     | Survey Earth Resources Observation and Science (USGS/EROS)"               |
|                                                                     | See Page 6 Line 13-15                                                     |
| "USGS Line 27, page 6: in the same to at the same"                  | "in the same" was modified to "at the same"                               |
|                                                                     |                                                                           |
|                                                                     | See Page 7, Line 19                                                       |
| "Line 10, page 7: year to years"                                    | The sentence has been modified as "Thirty thousand samples were collected |
|                                                                     | within four hurricane-prone zones in years 2003, 2004 and 2005,           |
|                                                                     | respectively."                                                            |
|                                                                     | See Page 8, Line 1-2                                                      |
| "Line 25, page 7: on to on the"                                     | "on" was modified to "on the"                                             |
|                                                                     |                                                                           |
|                                                                     | See Page 8, Line 16                                                       |
| "Line 27, page 7: upwards or downwards to upward or downward trend" | "upwards or downwards" was modified to "upward or downward"               |
|                                                                     | See Page 8, Line 18                                                       |
| "Line 3, page 10: a R2 to an R2"                                    | "a R2" was modified to "an R2"                                            |
|                                                                     |                                                                           |
|                                                                     | See Page 10, Line 11                                                      |
| "Line 7, page 12: year to years"                                    | "year" was modified to "years"                                            |
|                                                                     | See Page 13. Line 7                                                       |
| "Line13, page 12: decreased to decrease by"                         | "decrease" was modified to "decrease by"                                  |
|                                                                     |                                                                           |
|                                                                     | See Page 13, Line 12-13                                                   |
| "Line 9, page 12: City to the city"                                 | "City" was modified to "city"                                             |
|                                                                     | See Page 13, Line 9                                                       |
|                                                                     |                                                                           |

| "Line 17, page 12: Houston reveals dramatic to Houston has a"        | We modified this sentence to "Houston (Fig. 5e), for instance, has dramatically increased its human settlement." |
|----------------------------------------------------------------------|------------------------------------------------------------------------------------------------------------------|
|                                                                      | See Page 12 Line 17                                                                                              |
|                                                                      | See Fage 15, Line 17                                                                                             |
| "Line 3, page 13: matches to match"                                  | "matches" was modified to "match"                                                                                |
|                                                                      | See Page 14, Line 3                                                                                              |
| "Line 8, page 13: affect to affects"                                 | "affect" was modified to "affects"                                                                               |
|                                                                      | See Page 14 Line 8                                                                                               |
|                                                                      | See Fage 14, Line 8                                                                                              |
| "Line 1, page 15: percentage of area to percent area"                | "percentage of area" was modified to "percent area"                                                              |
|                                                                      | See Page 16, Figure 6 caption                                                                                    |
| "Line6 page 15. Zonal statistic to Zonal statistics: in to for"      | We modified this sentence to "Zonal statistics were also summarized for the                                      |
| Lineo, page 101 Lonai stansne to Lonai stansnes, in to jot           | four hurricane-prone zones (Table 3)."                                                                           |
|                                                                      |                                                                                                                  |
|                                                                      | Page 16, Line 6-7                                                                                                |
| "Line 7, page 15: Please rephrase "The net increase area calculates" | We modified this sentence to "The net increase area is defined as the area                                       |
|                                                                      | difference between pixels with a significant increasing and decreasing trend."                                   |
|                                                                      |                                                                                                                  |
|                                                                      | See Page 16, Line 7-8                                                                                            |
| "Line 11, page 15: significant to a significant"              | "significant" was modified to "a significant"                                                                    |
|                                                                      |                                                                                                                  |
|                                                                      | See Page 16, Line 11                                                                                             |
| "Line 12, page 18: Please rephrase "The three two years (2016-2018)" | We modified this sentence to "The last three years (2016-2018) have seen                                         |
|                                                                      | consecutive above-average damaging Atlantic hurricane seasons."                                                  |
|                                                                      | See Page 19, Line 12-13                                                                                          |
| "Line 18, page 19. Rephrase "an areal percentage of 4,22% in Zone 1  | We modified this sentence to "Via trend analysis, 4.22% of the area in Zone 1                                    |
| experienced significant increase in"                                 | experienced a significant increase in settlement intensity followed by 2 34%                                     |
|                                                                      | in Zone 2, 2.08% in Zone 3 and 1.65% in Zone 4, revealing higher pressure of                                     |
|                                                                      | human settlement and thus impacts from hurricanes in the frontmost coastal                                       |
|                                                                      | areas."                                                                                                          |
|                                                                      |                                                                                                                  |
|                                                                      | See Page 20, Line 17-20                                                                                          |
|                                                                      |                                                                                                                  |

**Response to RC 2**

Dear second referee,

Thanks for providing your valuable insights during our open discussion. We are very glad that you are satisfied with our explanation on your major concerns. In this letter, we summarized our discussion and provided details of our revision in a table.

| Comments                                                                         | Responses                                                                                                                                                                                                                                                                                                                                                                                                                                                                                                                                                                                                                                                                                                                                                                                                                                                                                                                                                                                                                                                                                                                                                                                                                                                                                                                                                                                                                                                                                                                                                                                                                                                                                                                                                                                                                                                                                                                                                                                                                                                                                                                          |
|----------------------------------------------------------------------------------|------------------------------------------------------------------------------------------------------------------------------------------------------------------------------------------------------------------------------------------------------------------------------------------------------------------------------------------------------------------------------------------------------------------------------------------------------------------------------------------------------------------------------------------------------------------------------------------------------------------------------------------------------------------------------------------------------------------------------------------------------------------------------------------------------------------------------------------------------------------------------------------------------------------------------------------------------------------------------------------------------------------------------------------------------------------------------------------------------------------------------------------------------------------------------------------------------------------------------------------------------------------------------------------------------------------------------------------------------------------------------------------------------------------------------------------------------------------------------------------------------------------------------------------------------------------------------------------------------------------------------------------------------------------------------------------------------------------------------------------------------------------------------------------------------------------------------------------------------------------------------------------------------------------------------------------------------------------------------------------------------------------------------------------------------------------------------------------------------------------------------------|
| "To what extent the percentage of pixels with VANUI>0 is representative of an    | Thanks for pointing out the issue. We totally understand your concern. Here are                                                                                                                                                                                                                                                                                                                                                                                                                                                                                                                                                                                                                                                                                                                                                                                                                                                                                                                                                                                                                                                                                                                                                                                                                                                                                                                                                                                                                                                                                                                                                                                                                                                                                                                                                                                                                                                                                                                                                                                                                                                    |
| increased land development? If we look at the case of Philadelphia (Fig. 5b) the | the seasons we'd like to present to you.                                                                                                                                                                                                                                                                                                                                                                                                                                                                                                                                                                                                                                                                                                                                                                                                                                                                                                                                                                                                                                                                                                                                                                                                                                                                                                                                                                                                                                                                                                                                                                                                                                                                                                                                                                                                                                                                                                                                                                                                                                                                                           |
| percentage does not significantly change from 1992 to 2013. What is changing     |                                                                                                                                                                                                                                                                                                                                                                                                                                                                                                                                                                                                                                                                                                                                                                                                                                                                                                                                                                                                                                                                                                                                                                                                                                                                                                                                                                                                                                                                                                                                                                                                                                                                                                                                                                                                                                                                                                                                                                                                                                                                                                                                    |
| here is the sum of VANUI values. Therefore, I strongly recommend to analyze      | Firstly, VANUI = $(1$ -vegetation) * light. We could claim that VANUI > 0                                                                                                                                                                                                                                                                                                                                                                                                                                                                                                                                                                                                                                                                                                                                                                                                                                                                                                                                                                                                                                                                                                                                                                                                                                                                                                                                                                                                                                                                                                                                                                                                                                                                                                                                                                                                                                                                                                                                                                                                                                                          |
| the sum of VANUI to check for land development. If the sum of VANUI does not     | represents "lights AND impervious surface", meaning that it has to satisfy two                                                                                                                                                                                                                                                                                                                                                                                                                                                                                                                                                                                                                                                                                                                                                                                                                                                                                                                                                                                                                                                                                                                                                                                                                                                                                                                                                                                                                                                                                                                                                                                                                                                                                                                                                                                                                                                                                                                                                                                                                                                     |
| confirm your previous findings, then your analysis presents a major flaw and     | requirements: 1) lights casted to a certain area and 2) that area has to have                                                                                                                                                                                                                                                                                                                                                                                                                                                                                                                                                                                                                                                                                                                                                                                                                                                                                                                                                                                                                                                                                                                                                                                                                                                                                                                                                                                                                                                                                                                                                                                                                                                                                                                                                                                                                                                                                                                                                                                                                                                      |
| cannot be accepted for publication."                                             | impervious surface. Traditional remote sensing (multispectral) that gauges                                                                                                                                                                                                                                                                                                                                                                                                                                                                                                                                                                                                                                                                                                                                                                                                                                                                                                                                                                                                                                                                                                                                                                                                                                                                                                                                                                                                                                                                                                                                                                                                                                                                                                                                                                                                                                                                                                                                                                                                                                                         |
|                                                                                  | urban expansion mainly consider the expansion of impervious surface. VANUI                                                                                                                                                                                                                                                                                                                                                                                                                                                                                                                                                                                                                                                                                                                                                                                                                                                                                                                                                                                                                                                                                                                                                                                                                                                                                                                                                                                                                                                                                                                                                                                                                                                                                                                                                                                                                                                                                                                                                                                                                                                         |
|                                                                                  | transcends those approaches by introducing another aspect of "lights". We                                                                                                                                                                                                                                                                                                                                                                                                                                                                                                                                                                                                                                                                                                                                                                                                                                                                                                                                                                                                                                                                                                                                                                                                                                                                                                                                                                                                                                                                                                                                                                                                                                                                                                                                                                                                                                                                                                                                                                                                                                                          |
|                                                                                  | believe that the evolvement of percentage of pixels ( $vANOI > 0$ ) during the 22-                                                                                                                                                                                                                                                                                                                                                                                                                                                                                                                                                                                                                                                                                                                                                                                                                                                                                                                                                                                                                                                                                                                                                                                                                                                                                                                                                                                                                                                                                                                                                                                                                                                                                                                                                                                                                                                                                                                                                                                                                                                 |
|                                                                                  | year investigated period musticates the field of the expansion in each zone. If in
Zone 1, there are 30% of pixels (VANUE $> 0$ ) in 1992 and 10 year later, there                                                                                                                                                                                                                                                                                                                                                                                                                                                                                                                                                                                                                                                                                                                                                                                                                                                                                                                                                                                                                                                                                                                                                                                                                                                                                                                                                                                                                                                                                                                                                                                                                                                                                                                                                                                                                                                                                                                                                              |
|                                                                                  | are 40% of pixels (VANUL > 0) in 2002 we believe the difference in the                                                                                                                                                                                                                                                                                                                                                                                                                                                                                                                                                                                                                                                                                                                                                                                                                                                                                                                                                                                                                                                                                                                                                                                                                                                                                                                                                                                                                                                                                                                                                                                                                                                                                                                                                                                                                                                                                                                                                                                                                                                             |
|                                                                                  | $\frac{1}{2002}$ , we believe the unreference in the intervence in the |
|                                                                                  | percentage mustates the expansion of number settement.                                                                                                                                                                                                                                                                                                                                                                                                                                                                                                                                                                                                                                                                                                                                                                                                                                                                                                                                                                                                                                                                                                                                                                                                                                                                                                                                                                                                                                                                                                                                                                                                                                                                                                                                                                                                                                                                                                                                                                                                                                                                             |
|                                                                                  | Secondly, due to the saturation problem of DMSP (value capped at 63), we are                                                                                                                                                                                                                                                                                                                                                                                                                                                                                                                                                                                                                                                                                                                                                                                                                                                                                                                                                                                                                                                                                                                                                                                                                                                                                                                                                                                                                                                                                                                                                                                                                                                                                                                                                                                                                                                                                                                                                                                                                                                       |
|                                                                                  | worried that the sum of VANUI for large-scale analysis might suffer from some                                                                                                                                                                                                                                                                                                                                                                                                                                                                                                                                                                                                                                                                                                                                                                                                                                                                                                                                                                                                                                                                                                                                                                                                                                                                                                                                                                                                                                                                                                                                                                                                                                                                                                                                                                                                                                                                                                                                                                                                                                                      |
|                                                                                  | uncertainties (we did the sum of VANUI for different MSAs in CONUS, please                                                                                                                                                                                                                                                                                                                                                                                                                                                                                                                                                                                                                                                                                                                                                                                                                                                                                                                                                                                                                                                                                                                                                                                                                                                                                                                                                                                                                                                                                                                                                                                                                                                                                                                                                                                                                                                                                                                                                                                                                                                         |
|                                                                                  | see table 4).                                                                                                                                                                                                                                                                                                                                                                                                                                                                                                                                                                                                                                                                                                                                                                                                                                                                                                                                                                                                                                                                                                                                                                                                                                                                                                                                                                                                                                                                                                                                                                                                                                                                                                                                                                                                                                                                                                                                                                                                                                                                                                                      |
|                                                                                  |                                                                                                                                                                                                                                                                                                                                                                                                                                                                                                                                                                                                                                                                                                                                                                                                                                                                                                                                                                                                                                                                                                                                                                                                                                                                                                                                                                                                                                                                                                                                                                                                                                                                                                                                                                                                                                                                                                                                                                                                                                                                                                                                    |
|                                                                                  | Thirdly, using the concept of "VANUI > a certain threshold" to perform regional                                                                                                                                                                                                                                                                                                                                                                                                                                                                                                                                                                                                                                                                                                                                                                                                                                                                                                                                                                                                                                                                                                                                                                                                                                                                                                                                                                                                                                                                                                                                                                                                                                                                                                                                                                                                                                                                                                                                                                                                                                                    |
|                                                                                  | analysis has been proved efficient by many (Li et al., 2016; Lu et al., 2018). In                                                                                                                                                                                                                                                                                                                                                                                                                                                                                                                                                                                                                                                                                                                                                                                                                                                                                                                                                                                                                                                                                                                                                                                                                                                                                                                                                                                                                                                                                                                                                                                                                                                                                                                                                                                                                                                                                                                                                                                                                                                  |
|                                                                                  | this study specifically, we set the threshold to 0.                                                                                                                                                                                                                                                                                                                                                                                                                                                                                                                                                                                                                                                                                                                                                                                                                                                                                                                                                                                                                                                                                                                                                                                                                                                                                                                                                                                                                                                                                                                                                                                                                                                                                                                                                                                                                                                                                                                                                                                                                                                                                |
|                                                                                  |                                                                                                                                                                                                                                                                                                                                                                                                                                                                                                                                                                                                                                                                                                                                                                                                                                                                                                                                                                                                                                                                                                                                                                                                                                                                                                                                                                                                                                                                                                                                                                                                                                                                                                                                                                                                                                                                                                                                                                                                                                                                                                                                    |
|                                                                                  | As for your concern, we confirmed that the trend of VANUI summation in
different games agrees well with what we have found using agreements of                                                                                                                                                                                                                                                                                                                                                                                                                                                                                                                                                                                                                                                                                                                                                                                                                                                                                                                                                                                                                                                                                                                                                                                                                                                                                                                                                                                                                                                                                                                                                                                                                                                                                                                                                                                                                                                                                                                                                                                  |
|                                                                                  | Unterent zones agrees wen with what we have found using percentage of VANUUS 0. We performed the statistics for Zone 1 and Zone 2 using sum of                                                                                                                                                                                                                                                                                                                                                                                                                                                                                                                                                                                                                                                                                                                                                                                                                                                                                                                                                                                                                                                                                                                                                                                                                                                                                                                                                                                                                                                                                                                                                                                                                                                                                                                                                                                                                                                                                                                                                                                     |
|                                                                                  | VANULO, we performed the same log trend while Zone 3 and Zone 4 avhibit no                                                                                                                                                                                                                                                                                                                                                                                                                                                                                                                                                                                                                                                                                                                                                                                                                                                                                                                                                                                                                                                                                                                                                                                                                                                                                                                                                                                                                                                                                                                                                                                                                                                                                                                                                                                                                                                                                                                                                                                                                                                         |
|                                                                                  | significant trend:                                                                                                                                                                                                                                                                                                                                                                                                                                                                                                                                                                                                                                                                                                                                                                                                                                                                                                                                                                                                                                                                                                                                                                                                                                                                                                                                                                                                                                                                                                                                                                                                                                                                                                                                                                                                                                                                                                                                                                                                                                                                                                                 |
|                                                                                  | significant ticht.                                                                                                                                                                                                                                                                                                                                                                                                                                                                                                                                                                                                                                                                                                                                                                                                                                                                                                                                                                                                                                                                                                                                                                                                                                                                                                                                                                                                                                                                                                                                                                                                                                                                                                                                                                                                                                                                                                                                                                                                                                                                                                                 |

|                                                                                                                                                                                                                                                                                                                                                                                                                                                                                                 | 60.0k -                                                                                                                                                                                                                                                                                                                                                                                                                                                                                                                                                                                                                                                                                                                                                                                                                                                                            |
|-------------------------------------------------------------------------------------------------------------------------------------------------------------------------------------------------------------------------------------------------------------------------------------------------------------------------------------------------------------------------------------------------------------------------------------------------------------------------------------------------|------------------------------------------------------------------------------------------------------------------------------------------------------------------------------------------------------------------------------------------------------------------------------------------------------------------------------------------------------------------------------------------------------------------------------------------------------------------------------------------------------------------------------------------------------------------------------------------------------------------------------------------------------------------------------------------------------------------------------------------------------------------------------------------------------------------------------------------------------------------------------------|
|                                                                                                                                                                                                                                                                                                                                                                                                                                                                                                 | 60.0k
55.0k
50.0k
45.0k
40.0k                                                                                                                                                                                                                                                                                                                                                                                                                                                                                                                                                                                                                                                                                                                                                                                                                                          |
|                                                                                                                                                                                                                                                                                                                                                                                                                                                                                                 | 35.0k
1992 1995 1998 2001 2004 2007 2010 2013
Year
80.0k
75.0k
Zone 2                                                                                                                                                                                                                                                                                                                                                                                                                                                                                                                                                                                                                                                                                                                                                                             |
|                                                                                                                                                                                                                                                                                                                                                                                                                                                                                                 | 100 70.0k
65.0k
55.0k
50.0k
50.0k                                                                                                                                                                                                                                                                                                                                                                                                                                                                                                                                                                                                                                                                                                                                                                                                                                      |
|                                                                                                                                                                                                                                                                                                                                                                                                                                                                                                 | 45.0k 1992 1995 1998 2001 2004 2007 2010 2013
Year                                                                                                                                                                                                                                                                                                                                                                                                                                                                                                                                                                                                                                                                                                                                                                                                                              |
| "Why sum slope to represent the rapidness of human settlement growth?***
Each pixel (1 km2) can have a maximum slope value of 1/22=0.045. By
summing slopes in a region, I assume that you can get an estimate of growth,
but this is proportional to the considered area (namely, one of the four
hurricane-prone zones), thus you cannot compare trends. To do this, you
should consider the average slope."                                                                   | Thanks for pointing the Theil-Sen slope issue. We totally agree with your opinion. Further, we'd like to explain our idea of using Theil-Sen slope. Mann-Kendall test only identifies pixels with significant trend (increasing or decreasing). It doesn't calculate the strength of the trend. After pixels are identified as having significant trend, Theil-Sen slope calculates how strong this trend is. By summing up all the Theil-Sen slopes in each zone, we can quantify the total increasing intensity in each zone. However, as you mentioned, this is proportional to the considered area. To solve this problem, we divided the summation of all the Theil-Sen slopes by the size of the zone. This is why (in Table 3) we added the column "Sum of Theil-Sen slope per 100,000 $km^2$ ". We are calculating the summation of Theil-Sen slope per unit in each zone. |
| "To what extent sum of slopes in Table 3 are significant? I do have serious
concerns related to the slope values reported in Table 3. If we assume that the
maximum slope per pixel is 0.045 and in zone 1 there are 312,453 pixels (all
starting from 0 in 1992 and reaching 1 in 2013), then ideally the maximum
sum value would be 14,060. How is this value related to 9.02 (per 100,000
km2)? Please elaborate more on this, to prove the significance of slope
values." | Please let us explain our methodology (Mann-Kendall + Theil-Sen slope) in
this study. Mann-Kendall identifies pixels with significant trend, and Theil-
Sen further calculates the slope of those pixels that have been identified . As
you pointed out, Zone 1 has 312,453 pixels. However, pixels with significant
increasing trend only occupy 4.22% (Table 3), meaning that the we are only
calculating the summation of Theil-Sen slope for around 13,185 pixels. Within
those 13,185 pixels, only a small amount of pixels have increased from 0 (no                                                                                                                                                                                                                                                                                                |

|                                                                                                                                                                                                                                                                                                                                                                                                                                                                                                                                                                                                                                                                                                                                                                                                                                                                                                                                                                                                                                                                                                                                                                                                                                                                                                                                                                                                                                                                                                                                                                                                                                                                                                                                                                                                                                                                                                                                                                                                                                                                                                                                | urban at all) to 1 (fully urbanized). If, all the pixels in Zone 1 have max Theil-
Sen slope (0.045), it should have a total of 593.3 (13,185 * 0.045), However, |
|--------------------------------------------------------------------------------------------------------------------------------------------------------------------------------------------------------------------------------------------------------------------------------------------------------------------------------------------------------------------------------------------------------------------------------------------------------------------------------------------------------------------------------------------------------------------------------------------------------------------------------------------------------------------------------------------------------------------------------------------------------------------------------------------------------------------------------------------------------------------------------------------------------------------------------------------------------------------------------------------------------------------------------------------------------------------------------------------------------------------------------------------------------------------------------------------------------------------------------------------------------------------------------------------------------------------------------------------------------------------------------------------------------------------------------------------------------------------------------------------------------------------------------------------------------------------------------------------------------------------------------------------------------------------------------------------------------------------------------------------------------------------------------------------------------------------------------------------------------------------------------------------------------------------------------------------------------------------------------------------------------------------------------------------------------------------------------------------------------------------------------|---------------------------------------------------------------------------------------------------------------------------------------------------------------------|
|                                                                                                                                                                                                                                                                                                                                                                                                                                                                                                                                                                                                                                                                                                                                                                                                                                                                                                                                                                                                                                                                                                                                                                                                                                                                                                                                                                                                                                                                                                                                                                                                                                                                                                                                                                                                                                                                                                                                                                                                                                                                                                                                | the total Theil-Sen slope in Zone 1 is around 28, meaning that pixels                                                                                               |
|                                                                                                                                                                                                                                                                                                                                                                                                                                                                                                                                                                                                                                                                                                                                                                                                                                                                                                                                                                                                                                                                                                                                                                                                                                                                                                                                                                                                                                                                                                                                                                                                                                                                                                                                                                                                                                                                                                                                                                                                                                                                                                                                | increasing from 0 to 1 occupy only a small percentage, which makes sense as                                                                                         |
|                                                                                                                                                                                                                                                                                                                                                                                                                                                                                                                                                                                                                                                                                                                                                                                                                                                                                                                                                                                                                                                                                                                                                                                                                                                                                                                                                                                                                                                                                                                                                                                                                                                                                                                                                                                                                                                                                                                                                                                                                                                                                                                                | areas that transform from pure rural (0) to pure urban (1) are limited in a                                                                                         |
|                                                                                                                                                                                                                                                                                                                                                                                                                                                                                                                                                                                                                                                                                                                                                                                                                                                                                                                                                                                                                                                                                                                                                                                                                                                                                                                                                                                                                                                                                                                                                                                                                                                                                                                                                                                                                                                                                                                                                                                                                                                                                                                                | developed country like U.S. In our perspective, the significance in Table 3 is                                                                                      |
|                                                                                                                                                                                                                                                                                                                                                                                                                                                                                                                                                                                                                                                                                                                                                                                                                                                                                                                                                                                                                                                                                                                                                                                                                                                                                                                                                                                                                                                                                                                                                                                                                                                                                                                                                                                                                                                                                                                                                                                                                                                                                                                                | not about the absolute value of "sum of Theil-Sen slope", but the comparison                                                                                        |
|                                                                                                                                                                                                                                                                                                                                                                                                                                                                                                                                                                                                                                                                                                                                                                                                                                                                                                                                                                                                                                                                                                                                                                                                                                                                                                                                                                                                                                                                                                                                                                                                                                                                                                                                                                                                                                                                                                                                                                                                                                                                                                                                | among different zones. After Mann-Kendall, as you may notice, the percentage                                                                                        |
|                                                                                                                                                                                                                                                                                                                                                                                                                                                                                                                                                                                                                                                                                                                                                                                                                                                                                                                                                                                                                                                                                                                                                                                                                                                                                                                                                                                                                                                                                                                                                                                                                                                                                                                                                                                                                                                                                                                                                                                                                                                                                                                                | of pixels with significant trend varies a lot in different zones (4.22% in Zone 1.                                                                                  |
|                                                                                                                                                                                                                                                                                                                                                                                                                                                                                                                                                                                                                                                                                                                                                                                                                                                                                                                                                                                                                                                                                                                                                                                                                                                                                                                                                                                                                                                                                                                                                                                                                                                                                                                                                                                                                                                                                                                                                                                                                                                                                                                                | nearly doubled from 2.34% in Zone 2). This statistic, however, only illustrates                                                                                     |
|                                                                                                                                                                                                                                                                                                                                                                                                                                                                                                                                                                                                                                                                                                                                                                                                                                                                                                                                                                                                                                                                                                                                                                                                                                                                                                                                                                                                                                                                                                                                                                                                                                                                                                                                                                                                                                                                                                                                                                                                                                                                                                                                | the coverage of significant pixels in different zones. It ignores the intensity of                                                                                  |
|                                                                                                                                                                                                                                                                                                                                                                                                                                                                                                                                                                                                                                                                                                                                                                                                                                                                                                                                                                                                                                                                                                                                                                                                                                                                                                                                                                                                                                                                                                                                                                                                                                                                                                                                                                                                                                                                                                                                                                                                                                                                                                                                | the increase. So we added the summation of Theil-Sen slope in order to                                                                                              |
|                                                                                                                                                                                                                                                                                                                                                                                                                                                                                                                                                                                                                                                                                                                                                                                                                                                                                                                                                                                                                                                                                                                                                                                                                                                                                                                                                                                                                                                                                                                                                                                                                                                                                                                                                                                                                                                                                                                                                                                                                                                                                                                                | capture the "intensity" of this increase. As we expected, Zone 1 have not only                                                                                      |
|                                                                                                                                                                                                                                                                                                                                                                                                                                                                                                                                                                                                                                                                                                                                                                                                                                                                                                                                                                                                                                                                                                                                                                                                                                                                                                                                                                                                                                                                                                                                                                                                                                                                                                                                                                                                                                                                                                                                                                                                                                                                                                                                | the highest percentage of significant pixels but also the highest intensity of this                                                                                 |
|                                                                                                                                                                                                                                                                                                                                                                                                                                                                                                                                                                                                                                                                                                                                                                                                                                                                                                                                                                                                                                                                                                                                                                                                                                                                                                                                                                                                                                                                                                                                                                                                                                                                                                                                                                                                                                                                                                                                                                                                                                                                                                                                | increase. In our revised manuscript, we illustrated that "As Table 3 suggests,                                                                                      |
|                                                                                                                                                                                                                                                                                                                                                                                                                                                                                                                                                                                                                                                                                                                                                                                                                                                                                                                                                                                                                                                                                                                                                                                                                                                                                                                                                                                                                                                                                                                                                                                                                                                                                                                                                                                                                                                                                                                                                                                                                                                                                                                                | 4.22% of the area in Zone 1 experienced a significant increase in human                                                                                             |
|                                                                                                                                                                                                                                                                                                                                                                                                                                                                                                                                                                                                                                                                                                                                                                                                                                                                                                                                                                                                                                                                                                                                                                                                                                                                                                                                                                                                                                                                                                                                                                                                                                                                                                                                                                                                                                                                                                                                                                                                                                                                                                                                | settlement, followed by 2.34% in Zone 2, 2.08% in Zone 3 and 1.65% in Zone                                                                                          |
|                                                                                                                                                                                                                                                                                                                                                                                                                                                                                                                                                                                                                                                                                                                                                                                                                                                                                                                                                                                                                                                                                                                                                                                                                                                                                                                                                                                                                                                                                                                                                                                                                                                                                                                                                                                                                                                                                                                                                                                                                                                                                                                                | 4. The statistics above suggest a noticeably positive relationship between the                                                                                      |
|                                                                                                                                                                                                                                                                                                                                                                                                                                                                                                                                                                                                                                                                                                                                                                                                                                                                                                                                                                                                                                                                                                                                                                                                                                                                                                                                                                                                                                                                                                                                                                                                                                                                                                                                                                                                                                                                                                                                                                                                                                                                                                                                | hurricane proneness of each zone and the percent area with a significant                                                                                            |
|                                                                                                                                                                                                                                                                                                                                                                                                                                                                                                                                                                                                                                                                                                                                                                                                                                                                                                                                                                                                                                                                                                                                                                                                                                                                                                                                                                                                                                                                                                                                                                                                                                                                                                                                                                                                                                                                                                                                                                                                                                                                                                                                | increase in settlement. The sum of Theil-Sen slope, on the other hand,                                                                                              |
|                                                                                                                                                                                                                                                                                                                                                                                                                                                                                                                                                                                                                                                                                                                                                                                                                                                                                                                                                                                                                                                                                                                                                                                                                                                                                                                                                                                                                                                                                                                                                                                                                                                                                                                                                                                                                                                                                                                                                                                                                                                                                                                                | established the relationship between the hurricane proneness and the increase                                                                                       |
|                                                                                                                                                                                                                                                                                                                                                                                                                                                                                                                                                                                                                                                                                                                                                                                                                                                                                                                                                                                                                                                                                                                                                                                                                                                                                                                                                                                                                                                                                                                                                                                                                                                                                                                                                                                                                                                                                                                                                                                                                                                                                                                                | rate of settlement in each zone. Zone 1 receives the most hurricane hits but has                                                                                    |
|                                                                                                                                                                                                                                                                                                                                                                                                                                                                                                                                                                                                                                                                                                                                                                                                                                                                                                                                                                                                                                                                                                                                                                                                                                                                                                                                                                                                                                                                                                                                                                                                                                                                                                                                                                                                                                                                                                                                                                                                                                                                                                                                | the strongest increase of settlement intensity, followed by Zone 2, Zone 3 and                                                                                      |
|                                                                                                                                                                                                                                                                                                                                                                                                                                                                                                                                                                                                                                                                                                                                                                                                                                                                                                                                                                                                                                                                                                                                                                                                                                                                                                                                                                                                                                                                                                                                                                                                                                                                                                                                                                                                                                                                                                                                                                                                                                                                                                                                | Zone 4."                                                                                                                                                            |
|                                                                                                                                                                                                                                                                                                                                                                                                                                                                                                                                                                                                                                                                                                                                                                                                                                                                                                                                                                                                                                                                                                                                                                                                                                                                                                                                                                                                                                                                                                                                                                                                                                                                                                                                                                                                                                                                                                                                                                                                                                                                                                                                | See Page 16, Line 9-14                                                                                                                                              |
| Page 5 L. 13: "resampled to the 1 km pixel size". NTL are already                                                                                                                                                                                                                                                                                                                                                                                                                                                                                                                                                                                                                                                                                                                                                                                                                                                                                                                                                                                                                                                                                                                                                                                                                                                                                                                                                                                                                                                                                                                                                                                                                                                                                                                                                                                                                                                                                                                                                                                                                                                              | DMSP NTL series has a resolution of 30 arc second, which transforms to around                                                                                       |
| at 1 km resolution.                                                                                                                                                                                                                                                                                                                                                                                                                                                                                                                                                                                                                                                                                                                                                                                                                                                                                                                                                                                                                                                                                                                                                                                                                                                                                                                                                                                                                                                                                                                                                                                                                                                                                                                                                                                                                                                                                                                                                                                                                                                                                                            | 1 km at the equator. However, the pixel size of raw DMSP data varies a lot,                                                                                         |
|                                                                                                                                                                                                                                                                                                                                                                                                                                                                                                                                                                                                                                                                                                                                                                                                                                                                                                                                                                                                                                                                                                                                                                                                                                                                                                                                                                                                                                                                                                                                                                                                                                                                                                                                                                                                                                                                                                                                                                                                                                                                                                                                | given different latitudes. In this study, we resampled both NTL and NDVI into                                                                                       |
|                                                                                                                                                                                                                                                                                                                                                                                                                                                                                                                                                                                                                                                                                                                                                                                                                                                                                                                                                                                                                                                                                                                                                                                                                                                                                                                                                                                                                                                                                                                                                                                                                                                                                                                                                                                                                                                                                                                                                                                                                                                                                                                                | 1 km grid using Lambert Azimuthal Equal-area projection.                                                                                                            |
| Page 6 L. 20-21: categorization of hurricane-prone zones: what is the                                                                                                                                                                                                                                                                                                                                                                                                                                                                                                                                                                                                                                                                                                                                                                                                                                                                                                                                                                                                                                                                                                                                                                                                                                                                                                                                                                                                                                                                                                                                                                                                                                                                                                                                                                                                                                                                                                                                                                                                                                                          | Thanks for pointing out the categorization issue. We used Jenks optimization                                                                                        |
| distribution of rho? You should consider (if not done yet) the frequency                                                                                                                                                                                                                                                                                                                                                                                                                                                                                                                                                                                                                                                                                                                                                                                                                                                                                                                                                                                                                                                                                                                                                                                                                                                                                                                                                                                                                                                                                                                                                                                                                                                                                                                                                                                                                                                                                                                                                                                                                                                       | method (also called Jenks natural breaks classification method) to determine                                                                                        |
| distribution of rho values to categorize zones.                                                                                                                                                                                                                                                                                                                                                                                                                                                                                                                                                                                                                                                                                                                                                                                                                                                                                                                                                                                                                                                                                                                                                                                                                                                                                                                                                                                                                                                                                                                                                                                                                                                                                                                                                                                                                                                                                                                                                                                                                                                                                | the arrangement of values into different class. The values in the sentence                                                                                          |
|                                                                                                                                                                                                                                                                                                                                                                                                                                                                                                                                                                                                                                                                                                                                                                                                                                                                                                                                                                                                                                                                                                                                                                                                                                                                                                                                                                                                                                                                                                                                                                                                                                                                                                                                                                                                                                                                                                                                                                                                                                                                                                                                | "Zone 4 (0-0.2), Zone 3 (0.2-0.5), Zone 2 (0.5-0.7) and Zone 1 (0.7-1.0)" are                                                                                       |
|                                                                                                                                                                                                                                                                                                                                                                                                                                                                                                                                                                                                                                                                                                                                                                                                                                                                                                                                                                                                                                                                                                                                                                                                                                                                                                                                                                                                                                                                                                                                                                                                                                                                                                                                                                                                                                                                                                                                                                                                                                                                                                                                | the rounded thresholds defined by Natural Jenks. We didn't include this                                                                                             |
|                                                                                                                                                                                                                                                                                                                                                                                                                                                                                                                                                                                                                                                                                                                                                                                                                                                                                                                                                                                                                                                                                                                                                                                                                                                                                                                                                                                                                                                                                                                                                                                                                                                                                                                                                                                                                                                                                                                                                                                                                                                                                                                                | information in the manuscript because we believe it is trivial compared to the                                                                                      |
|                                                                                                                                                                                                                                                                                                                                                                                                                                                                                                                                                                                                                                                                                                                                                                                                                                                                                                                                                                                                                                                                                                                                                                                                                                                                                                                                                                                                                                                                                                                                                                                                                                                                                                                                                                                                                                                                                                                                                                                                                                                                                                                                | entire workflow. Spearman's rho measures the strength of association between                                                                                        |
|                                                                                                                                                                                                                                                                                                                                                                                                                                                                                                                                                                                                                                                                                                                                                                                                                                                                                                                                                                                                                                                                                                                                                                                                                                                                                                                                                                                                                                                                                                                                                                                                                                                                                                                                                                                                                                                                                                                                                                                                                                                                                                                                | ranked variables. Since our density estimation is continuous, we believe                                                                                            |
|                                                                                                                                                                                                                                                                                                                                                                                                                                                                                                                                                                                                                                                                                                                                                                                                                                                                                                                                                                                                                                                                                                                                                                                                                                                                                                                                                                                                                                                                                                                                                                                                                                                                                                                                                                                                                                                                                                                                                                                                                                                                                                                                | Inatural Jenks method is more valid in this case. We appreciate your                                                                                                |
| "I 22.25. "Coming as the sectore as the total design of the sector of th | Understanding.                                                                                                                                                      |
| L. 25-25: "Serving as the reference site in that study [Elvidge et al. 2009]".                                                                                                                                                                                                                                                                                                                                                                                                                                                                                                                                                                                                                                                                                                                                                                                                                                                                                                                                                                                                                                                                                                                                                                                                                                                                                                                                                                                                                                                                                                                                                                                                                                                                                                                                                                                                                                                                                                                                                                                                                                                 | we sincerely apologize for the misuse of reference in our statement. Elvidge et                                                                                     |
| Elvidge et al 2009 considered Sicily as the reference site to perform                                                                                                                                                                                                                                                                                                                                                                                                                                                                                                                                                                                                                                                                                                                                                                                                                                                                                                                                                                                                                                                                                                                                                                                                                                                                                                                                                                                                                                                                                                                                                                                                                                                                                                                                                                                                                                                                                                                                                                                                                                                          | al. (2009) did consider Sicily as reference site. In Hsu et al. (2015), they stated                                                                                 |

| intercalibration, not Los Angeles and the City of San Diego, as you stated in             | that "Los Angeles was taken as the reference for the Radiance Calibrated             |
|-------------------------------------------------------------------------------------------|--------------------------------------------------------------------------------------|
| vour manuscript."                                                                         | products for two reasons. First, Los Angeles has long been a mature metropolis       |
|                                                                                           | and the light change is negligible. Second being a metropolis it can provide         |
|                                                                                           | samples with high DNs from the city center, as well as low DNs from the              |
|                                                                                           | subjust an area". We have revised this centeries as "Semine as the reference site    |
|                                                                                           | suburban area . We have revised this sentence as Serving as the reference site       |
|                                                                                           | (Fig. 3a), the geographic area of metropolitan Los Angeles and City of San           |
|                                                                                           | Diego, CA maintains high conformity of NTL values throughout the 22-year             |
|                                                                                           | period (Kyba et al., 2017; Hsu et al., 2015), which satisfies the "pseudo            |
|                                                                                           | invariant" rule for calibration site selection (Elvidge et al., 2009)."              |
|                                                                                           |                                                                                      |
|                                                                                           | See Page 7, Line 15-18                                                               |
| "L. 30: Why do you calibrate DN values if you employ NDVI values? Zhang et                | In the study of Zhang et al. (2013), he proposed the calculation of VANUI but        |
| al 2013 use original DN values. Please justify this."                                     | he did not focus on forming a time series. Since our NDVI were derived from          |
|                                                                                           | two different satellites, we think it is valid to perform a calibration for two      |
|                                                                                           | NDVI products as it is reported that NDVI for AVHRR and MODIS are                    |
|                                                                                           | different in a minor way (Gallo et al., 2004).                                       |
| "Page 7 L. 10: 30,000 samples – are 30,000 samples per year or 10,000                     | Sorry for the confusion this sentence might cause. It is 30,000 samples per year.    |
| samples per year? Is this representative of the range of NDVI values within the           | We have modified the sentence as "Thirty thousand samples were collected             |
| study area? If I am correct you examine 0.4% (if 30k in total) or 1.2% (if 30k            | within four hurricane-prone zones in years 2003, 2004 and 2005, respectively."       |
| ner year) of nivel to intercalibrate NDVI values "                                        | In terms of the representativeness, we believe a total of 90,000 should be enough    |
|                                                                                           | for a simple linear collibration. The result of $D^2 = 0.024$ in the collibration of |
|                                                                                           | for a simple linear caloration. The result of $R = 0.954$ in the caloration of       |
|                                                                                           | AVHRR and MODIS proves that the products of those two satellites are very            |
|                                                                                           | similar. Many studies have compared and calibrated NDVI, but their sample            |
|                                                                                           | sizes are mainly within thousands (Beck, et al., 2011) or tens of thousands (Gallo   |
|                                                                                           | et al., 2004).                                                                       |
|                                                                                           |                                                                                      |
|                                                                                           | See Page 8, Line 1-2                                                                 |
| " Eq. 5: " $k$ -1" should be " $k$ =1". " $j$ - $k$ +1" should be " $j$ = $k$ +1"" | "k-1" was modified to "k=1" and "j-k+1" was modified to "j=k+1"                      |
|                                                                                           | See Page 8 Equation 5                                                                |
| "Page & Eq. 6: "n 1" should be "n=1""                                                     | "n 1" was modified to "n=1"                                                          |
| $1 uge \circ Eq. \circ. p^{-1}$ should be $p^{-1}$                                        | p-1 was mounted to p-1                                                               |
|                                                                                           | See Page 8, Equation 6                                                               |
| "L 9: add the meaning of $Z=0$ in this sentence"                                          | Thank you for pointing out the missing explanation of $Z = 0$ . We have revised      |
|                                                                                           | the sentence as "The Z value in Eq.7 represents the monotonic tendency of a          |
|                                                                                           | time series A positive 7 indicates an increasing trend, while a negative 7           |
|                                                                                           | indicates a decreasing one. A stable trend exists when the value of 7 equals 0."     |
|                                                                                           | indicates a decreasing one. At studie active when the value of Z equals 0.           |
|                                                                                           | See Page 9, Line 2-3                                                                 |
| "L. 24: Fig 2a instead of Fig 1a"                                                         | "Fig 1a" was modified to "Fig 2a"                                                    |
|                                                                                           |                                                                                      |
|                                                                                           | See Pagey, Line 18-19                                                                |

| "Page 9 L. 1-4: Hew Hampshire is missing. Please add it to the list."                                                               | We apologize for the missing New Hampshire in our list. This sentence has
been revised as "The study area contains all U.S. states covered in the
hurricane-prone zones (Fig. 2c): Maine, Massachusetts, New Jersey, New
York, North Carolina, New Hampshire, Pennsylvania, Rhode Island,
Tennessee, Texas, Maryland, Alabama, Arkansas, Connecticut, Delaware, DC,
Florida, Georgia, Kentucky, Louisiana, Mississippi, South Carolina, Vermont,
Virginia and West Virginia."
See Page 9, Line 21-24                      |
|-------------------------------------------------------------------------------------------------------------------------------------|-

---

## Author Response (AR2)

**Technical Correction**

Dear Editor and reviewers:

We corrected our definition of "Hurricane" in the Introduction session. This sentence has been revised as:

**Hurricane, a specific type of tropical cyclone with wind speed of 74 miles per hour (mph) or higher, is one of the most devastating natural hazards in the world and is recurring more frequently than ever in coastal areas (Vecchi and Knutson, 2018).**

Specifically, in this correction we updated the affiliation and contact information for one of our authors, per his request. The authors and affiliation information are attached here:

**Xiao Huang[1], Cuizhen Wang[1], Junyu Lu[2]**

**[1]Department of Geography, University of South Carolina, Columbia, 29208, U.S.A**
**[2]School of Community Resources and Development, Arizona State University, Phoenix, 85004, U.S.A**

We also confirmed our funding source and modified the "Financial support" session as:

***Financial support.*** **This study is partially supported by a 2018 ASPIRE Grant and a 2018 SPARC Graduate Research Grant from the Office of the Vice Program, University of South Carolina.**

We appreciate your time.